

# Many-body chaos and anomalous diffusion across thermal phase transitions in two dimensions

**Sibaram Ruidas and Sumilan Banerjee⋆**

Centre for Condensed Matter Theory, Department of Physics, Indian Institute of Science, Bangalore 560012, India

⋆ sumilan@iisc.ac.in

## Abstract

Chaos is an important characterization of classical dynamical systems. How is chaos linked to the long-time dynamics of collective modes across phases and phase transitions? We address this by studying chaos across Ising and Kosterlitz-Thouless transitions in classical XXZ model. We show that spatio-temporal chaotic properties have crossovers across the transitions and distinct temperature dependence in the high and low-temperature phases which show normal and anomalous diffusions, respectively. Our results also provide new insights into the dynamics of interacting quantum systems in the semiclassical limit.

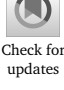

# 1   Introduction

Chaotic systems are described by a growth rate, the maximum Lyapunov exponent $\lambda_L$ ($> 0$), of perturbation to the initial condition. In recent years, a *quantum* Lyapunov exponent, and a butterfly velocity $v_B$ for ballistic spread of local perturbation, computed from the out-of-time-order commutator (OTOC) have emerged as important measures for chaos in quantum many-body systems having some well-defined semiclassical limit [1–4]. In these quantum systems where the Lyapunov exponent can be extracted, it has been perceived as a rate for early-to-intermediate-time thermalization. Here, thermalization refers to the emergence of statistical mechanical description during dynamical evolution of the system. However, it is still controversial [5] whether chaos in isolated interacting classical systems is essential for thermalization.

The recent interest in chaos in quantum many-body systems stems from the proof of a remarkable upper bound, $2\pi k_B T/\hbar$, for $\lambda_L$ [6] and a relation, $D \sim v_B^2/\lambda_L$, between diffusion coefficient $D$, a quantity related to transport, and $\lambda_L$ and $v_B$ in certain strongly interacting systems [7]. Such bounds have been phenomenologically conjectured for transport scattering rate [8,9], or more conventional relaxation rates extracted from usual time-ordered correlation functions. However, these conjectured bounds have not been concretely established. As a result, even if we set aside the issue of actual role of chaos in thermalization, the temperature ($T$) dependence of $\lambda_L$, and other quantities related to chaos, can serve as a fundamental characterization of phases and phase transition as various chaotic fixed points, like certain non-Fermi liquids and Fermi liquids [10–13]. The former are highly chaotic with $\lambda_L \sim T$ at low temperature, whereas the weakly interacting Fermi liquids show $\lambda_L \sim T^2$, essentially dictated by quasiparticle decay rate. Motivated by these, here we ask whether many-body chaos can be used to classify phases and finite-$T$ phase transitions in classical systems with intrinsic dynamics. An example of such systems is interacting classical Heisenberg spins on a lattice with precession dynamics.

However, typically, chaos is a probe of the short- and intermediate-time behaviour. On the other hand, the long-time dynamical properties of interacting many-body systems, in symmetry broken and unbroken phases, and across phase transitions, are mostly characterized by the properties of the collective low-energy excitations, hydrodynamic and critical modes. How are the short-time chaotic properties of many-body systems related to their long-time dynamics? To address these questions, we look into the connection of chaos with transport, characterized in terms of usual dynamical spin-spin correlations in the spin system.

Classical many-body chaos under Poisson-bracket spin dynamics has been studied in some recent works for various spin models at high [14] and low temperatures [15], as well as across a spin-glass transition in a zero-dimensional spin model [16]. Also, there have been similar

studies in other dynamical models like Burgers hydrodynamics [17], and in the classical limit of a relativistic field theory [18]. There are many earlier studies of maximum Lyapunov exponent as well as full Lyapunov spectrum in spin models, molecular systems, and across classical phase transitions [19–30]. The conventional Lyapunov spectrum analysis, in principle, can reveal the spatiotemporal structure of chaos [31–33]. However, Lyapunov spectrum analysis typically makes the direct information about the spatial structure somewhat obscure. In this regard, a spatiotemporal *correlation function* like OTOC [14, 34] provides much more transparent way by explicitly revealing spatial spread of a local perturbation. Thus, in contrast to earlier works [19–30], using a classical version of OTOC [14], we establish the detailed temperature dependence of both temporal ($\lambda_{\rm L}$) and spatial ($\nu_B$) characteristics of chaos across two *classic* thermal phase transitions in two dimensions. The models are described by a microscopic spin dynamics that is directly connected with the quantum dynamics in the semiclassical limit. We show that the chaotic properties, in general, are rather impervious to the nature of transport, namely whether the system exhibits diffusion or anomalous diffusion.

## 2 Model, dynamics and OTOC

We study the classical XXZ model on a square lattice described by the Hamiltonian

$$\mathcal{H} = -\frac{J}{2} \sum_{r,\delta} \left( S_{\mathbf{r}}^x S_{\mathbf{r}+\delta}^x + S_{\mathbf{r}}^y S_{\mathbf{r}+\delta}^y + \Delta S_{\mathbf{r}}^z S_{\mathbf{r}+\delta}^z \right), \tag{1}$$

where $\mathbf{S_r} = (S_{\mathbf{r}}^x, S_{\mathbf{r}}^y, S_{\mathbf{r}}^z)$ are unit length spin vectors on lattice site $\mathbf{r}$ with total $N$ sites and $J$ is the coupling between spins on the nearest neighbor bonds along $\delta = \pm\hat{\mathbf{x}}, \pm\hat{\mathbf{y}}$ directions. The anisotropy $\Delta \geq 0$ can be varied to change the nature of the finite-$T$ phase transition. For example, the system undergoes a transition at a non-zero temperature $T_{\rm KT}$ in the Kosterlitz-Thouless (KT) [35] universality class for $\Delta < 1$ (*easy plane*), and an Ising transition for $\Delta > 1$ (*easy axis*), while the isotropic ($\Delta = 1$) point does not have any finite-$T$ transition. The chaotic properties of the isotropic point have been studied by Bilitewski et al. [36] in an independent work.

We study chaotic properties of the model in Eq.(1) using the classical OTOC [14], along with more conventional dynamical spin correlation function $\langle \mathbf{S_r}(t) \cdot \mathbf{S_{r'}}(0) \rangle$, for the Poisson bracket dynamics

$$\frac{d\mathbf{S_r}}{dt} = \{\mathbf{S_r}, \mathcal{H}\} = \mathbf{S_r} \times \mathbf{h_r}. \tag{2}$$

The Poisson bracket of two functions $f(\{\mathbf{S_r}\})$ and $g(\{\mathbf{S_r}\})$ is defined as $\{f, g\} = \sum_{\mathbf{r}ijk} \epsilon^{ijk} (\partial f / \partial S_{\mathbf{r}}^i)(\partial g / \partial S_{\mathbf{r}}^j) S_{\mathbf{r}}^k$, where $\epsilon^{ijk}$ Levi-Civita tensor with $i, j, k = x, y, z$. Here $\mathbf{h_r} = J \sum_{\delta} (S_{\mathbf{r}+\delta}^x \hat{\mathbf{x}} + S_{\mathbf{r}+\delta}^y \hat{\mathbf{y}} + \Delta S_{\mathbf{r}+\delta}^z \hat{\mathbf{z}})$ is the effective field on the spin at $\mathbf{r}$. For $\Delta \neq 1$, apart from total energy, the dynamics conserves $S_{\rm total}^z = \sum_{\mathbf{r}} S_{\mathbf{r}}^z$. This hydrodynamic mode is expected to lead to diffusive behaviour for dynamical spin correlation function at long times.

To characterize the chaotic properties of the model of Eq.(1), we use a classical version of OTOC, or the so-called *cross correlator* or *decorrelator*, introduced in ref. [14],

$$\mathcal{D}(\mathbf{r}, t) \equiv 1 - \langle \mathbf{S}_{a\mathbf{r}}(t) \cdot \mathbf{S}_{b\mathbf{r}}(t) \rangle, \tag{3}$$

where $a$ and $b$ denote two copies of the initial configuration ($t = 0$), with $b$ slightly perturbed from $a$ at $\mathbf{r} = 0$ such that $\mathbf{S}_{b\mathbf{r}}(0) = \mathbf{S}_{a\mathbf{r}}(0) + \delta\mathbf{S}_0 \delta_{\mathbf{r},0}$. The small perturbation, $|\delta\mathbf{S}_0| \approx \varepsilon$, is chosen to be orthogonal to $\mathbf{S}_{a0}(0)$, i.e. $\delta\mathbf{S}_0 \cdot \mathbf{S}_{a0}(0) = 0$. More specifically, following Ref. [14], we generate the initial configuration $\{\mathbf{S}_{b\mathbf{r}}(0)\}$ for replica $b$ by rotating $\mathbf{S}_{a\mathbf{0}}$ slightly about a unit

vector $\hat{\mathbf{n}} = (\hat{\mathbf{z}} \times \mathbf{S}_{a0})/|(\hat{\mathbf{z}} \times \mathbf{S}_{a0})|$ such that the perturbation at $t = 0$ becomes $\delta\mathbf{S_0} = \varepsilon(\hat{\mathbf{n}} \times \mathbf{S}_{a0})$. The perturbation only preserves the normalization of the spin $\mathbf{S}_{b0}$ at $\mathbf{r} = \mathbf{0}$ up to $\mathcal{O}(\varepsilon)$, i.e. $\mathbf{S}_{b0}^2 \simeq 1 + \mathcal{O}(\varepsilon^2)$. For the particular choice of perturbation, the connection of $\mathcal{D}(\mathbf{r}, t)$ with the quantum out-of-time-ordered commutator $\langle [\mathbf{S_r}(t), \hat{\mathbf{n}} \cdot \mathbf{S_0}(0)]^2 \rangle$ in the semi-classical limit, where commutator is replaced by Poisson bracket, has been discussed in Ref. [14]. We note that $\mathcal{D}(\mathbf{r} \neq \mathbf{0}, 0)$ identically zero, and, since $\delta\mathbf{S_0} \perp \mathbf{S}_{a0}$, $\mathcal{D}(\mathbf{0}, 0) = 0$ too as $\mathbf{S}_{a\mathbf{r}}(0) \cdot \mathbf{S}_{b\mathbf{r}}(0) = 1$ at any $\mathbf{r}$. Thus, $\mathcal{D}(\mathbf{r}, t)$ starts from zero for any $\mathbf{r}$ due to the special choice of the initial orthogonal perturbation. The averaging $\langle \ldots \rangle$ is over initial equilibrated spin configurations $\{\mathbf{S}_{a\mathbf{r}}(0)\}$ drawn from a thermal distribution $\propto e^{-\mathcal{H}(\{\mathbf{S}_{a\mathbf{r}}(0)\})/T}$ (Boltzmann constant $k_{\mathrm{B}} = 1$). Starting from the slightly different initial conditions as discussed above, the two copies are time evolved independently via spin-precession dynamics of Eq.(2). The classical OTOC $\mathcal{D}(\mathbf{r}, t)$ measures the amount of *de-correlation* at $(\mathbf{r}, t)$ between the configurations in the two replicas or the trajectories, which are almost completely correlated at $t = 0$. The classical OTOC differs at $\mathcal{O}(\varepsilon^2)$ for $\mathbf{r} = 0$, $t = 0$ from the more conventional measure of spatio-temporal divergence of two trajectories [37], $\langle (\delta\mathbf{S_r}(t))^2 \rangle = \langle (\mathbf{S}_{a\mathbf{r}}(t) - \mathbf{S}_{b\mathbf{r}}(t))^2 \rangle$, which we denote as *trajectory divergence* for brevity. Starting at $\langle (\delta\mathbf{S_0}(0))^2 \rangle \sim \mathcal{O}(\varepsilon^2)$ at $t = 0$, the trajectory divergence is expected to grow exponentially at $\mathbf{r} = 0$ as $\varepsilon^2 e^{2\lambda_{\mathrm{L}} t}$ over a Lyapunov time window $t \sim \lambda_{\mathrm{L}}^{-1} \ln \varepsilon^{-2}$ in a chaotic system. We show that $\langle (\delta\mathbf{S_r}(t))^2 \rangle$ and the classical OTOC of Eq.(3), both have an early-time regime $0 \leq t \leq t_0$, where $\langle (\delta\mathbf{S_0}(t))^2 \rangle$ and $\mathcal{D}(0, t)$ initially change non-exponentially to $\mathcal{O}(\varepsilon^2)$, and then grows exponentially for $t > t_0$. In fact, both $\langle (\delta\mathbf{S_0}(t))^2 \rangle$ and $\mathcal{D}(0, t)$ initially decrease before start increasing with a power-law time dependence till $t_0$. Nonetheless, $t_0$ is found to be closely connected with the chaos time scale $1/\lambda_{\mathrm{L}}$, as we discuss later.

The main motivations for studying the spatio-temporal OTOC [Eq.(3)] in the model of Eq.(1) are twofold. One, as stated in the introduction, is to dynamically characterize the thermodynamic phase diagram of a spin model with well-known two-dimensional (2d) phase transitions. The XXZ model allows to tune the relative contribution of various hydrodynamic, low-energy and critical modes in the dynamics by changing temperature and anisotropy, and thus to probe the potential role of these collective modes on chaos. The second motivation comes from the fact that the spin precession dynamics [Eq.(2)] can be obtained as a classical large-$S$ (spin) limit of the Heisenberg equation of motion for the quantum XXZ model. Hence, chaotic properties of such classical model can give useful insights even about the quantum model. The results from the classical dynamics could be particularly relevant near finite-temperature continuous phase transitions, where quantum effects for the dynamics are generally believed to be irrelevant [38] due to divergent length and time scales.

In quantum systems, truly chaotic behaviour, namely the exponential growth of OTOC [34], can only be observed certain large-$N$ models, e.g. Sachdev-Ye-Kitaev (SYK) and related models dual to black holes [1, 3, 4, 7, 10, 11, 39], other large-$N$ theories [40–42], and weakly interacting systems with semiclassical quasiparticle dynamics [43, 44]. In these models the exponential growth can be observed over a parametrically long time window between $t \sim \lambda_{\mathrm{L}}^{-1}$ and $\lambda_{\mathrm{L}}^{-1} \ln N$ or $\lambda_{\mathrm{L}}^{-1} \ln(1/\hbar)$ for large $N$ or the semiclassical ($\hbar \to 0$) limits, respectively. The large-$N$ models are either infinite range or have a large local Hilbert space. In contrast, short-range quantum models with finite local Hilbert space, and without any semiclassical limit, typically do not show any exponential growth regime in OTOC [34, 45, 46]. This lack of exponential growth is either simply due to the absence of chaotic growth or else due to very short, and thus unresolvable, temporal window of the growth. It is an outstanding unresolved question whether such quantum systems can show chaos. However, as shown in Ref. [47], the semiclassical limit, though sufficient, may not be a necessary condition to observe the exponential growth. In particular, even for a short-range quantum model with finite local Hilbert space and without any obvious semiclassical limit, the exponential growth may be ascertained through a suitably defined spatially integrated OTOC if $v_B/\lambda_{\mathrm{L}}\ell \gg 1$, where $\ell$ is a

microscopic length scale. Based on our calculations in the classical limit, we identify a possible temperature regime in the XXZ model where such a condition could be satisfied, and thus the exponential growth may be observed even in the quantum limit. Moreover, as remarked earlier, quantum effects typically become unimportant near finite-temperature continuous phase transitions. Thus, one can naively conjecture that even short-range quantum models with finite local Hilbert space may show chaos due to effective coarse graining of degrees of freedom near the transitions. However, chaos is only short and intermediate-time property and maybe unaffected by such critical coarse-graining at long time scale. Nevertheless, the exploration of this possibility will require the simulation of real-time dynamics of the quantum XXZ model across the 2d transitions and is beyond the scope of this paper. Here we only study the chaotic properties across the phase transition in the large-$S$ limit of the XXZ model.

In the context of the interrelation between chaos and the dynamics of collective mode, we particularly focus on the dependence of chaos on the nature of transport in the presences of conserved quantities. Interesting interplay between operator spreading characterized via OTOC and diffusion due to conserved modes have been explored in quantum systems [48,49], albeit in the toy models of random unitary circuits [50,51]. However, these toy models are non chaotic from the perspective of exponential growth of OTOC, though they can be classified as quantum chaotic based on other diagnostics, like entanglement growth [52]. For the chaotic quantum systems of strongly interacting diffusive metal [7,39] built from solvable large-$N$ SYK model, the OTOC exhibits exponential growth with a ballistic light cone, i.e. $\mathcal{D}(\mathbf{r}, t) \sim \exp[\lambda_{\mathrm{L}}(t - r/v_B)]$, with $v_B^2/\lambda_{\mathrm{L}}$ exactly equal to the energy diffusion constant. Moreover, in weakly-interacting diffusive metal, $\mathcal{D}(\mathbf{r}, t) \sim \exp[\lambda_{\mathrm{L}} t (1 - (r/v_B t)^2)]$ with charge diffusion constant $D = v_B^2/4\lambda_{\mathrm{L}}$. Similar relation between spin diffusion constant and $v_B^2/\lambda_{\mathrm{L}}$ has been deduced numerically in the interacting classical spin-liquid regime of a frustrated spin system [15]. We find the functional form, $\mathcal{D}(\mathbf{r}, t) \sim \exp[\lambda_{\mathrm{L}} t (1 - (r/v_B t)^{\nu})]$, with $\nu$ varying between 1 to 2 from low to high temperature, to be a good description for the OTOC close to the ballistic chaos front [34] for both easy-plane and easy-axis anisotropies. The relation $D = v_B^2/4\lambda_{\mathrm{L}}$ is violated either qualitatively or quantitatively even at high temperatures. Moreover, we find the evidence of anomalous diffusion at low and intermediate temperatures.

As mentioned in the introduction, there are many earlier studies [19,20,24,25,29] on Lyapunov exponent across phase transitions in classical lattice spin models. In the context of OTOC and spatio-temporal evolution of chaos across phase transitions, more recent results on OTOC in $O(N)$ models [18,41,42] are directly relevant for our work. As discussed later in detail, the temperature dependence and finite-size scaling of the butterfly speed $v_B$ close to the transitions could be related with dynamical critical exponent $z$ [53]. Ref. [18] performed numerical simulation of high-temperature classical dynamics of 2+1d relativistic quantum field theory with $O(1)$ order parameter. Unlike the large-$S$ classical limit in the quantum XXZ model, taking the classical limit of the 2+1d $O(1)$ field theory is somewhat more involved [18,54]. The high-temperature dynamics in 2+1d $O(1)$ model is relevant across finite-temperature 2d Ising phase transition in the model. The dynamics is relevant for 2d transverse field Ising model [55], but, it does not conserve the order parameter. This is unlike the XXZ dynamics in the easy-axis case considered here. The non-conserved order parameter dynamics in the $O(1)$ model falls in the *Model C* category among the dynamical universality classes [38] and has $z = 2$ [18,56,57] for 2d Ising transition. In contrast, the dynamics in Eq.(2) conserves the Ising order parameter, i.e. the $z$ component of spin for $\Delta > 1$, and expected to be in the *Model B or D* dynamical universality class with $z = 4 - \eta$ with anomalous exponent $\eta = 0.25$ for 2d Ising transition [35]. Refs. [41,42] obtained temperature dependence of $\lambda_{\mathrm{L}}$ and $v_B$ in the ordered and disordered phases, and in the quantum critical regime of 2+1d $O(N)$ model in the large $N$ approximation. The dynamics of the model for $N = 2$ is more appropriate for 2d quantum rotors and planar antiferromagnets [55] and the finite-temperature transition is

expected to have a dynamical exponent $z \approx 2$ [35, 56]. The large-$N$ approximation, unlike our direct numerical Monte Carlo and spin dynamics simulation in the XXZ model, cannot appropriately describe KT transition in the 2d $O(2)$ model. In contrast, the dynamics [Eq.(2)] in the ferromagnetic XXZ model for the easy-plane case $\Delta < 1$ is described by the *Model E* dynamics [35, 58, 59], where the Poisson bracket terms between planar spin components, i.e. the order parameter, and conserved $z$ component of spin are important. In this case, one expects a dynamical exponent $z = 1$ in 2d [58, 59]. We discuss these points further in the context of our results for $\nu_B(T)$.

# 3 Overview of the dynamical phase diagram of classical XXZ model: Chaos and dynamical correlations

Our main results are summarized schematically in a phase diagram in Fig.1. Before describing the results in detail, we give an overview of our main results below.

1. We show that $\lambda_L(T)$ has a crossover across both KT and Ising transitions, clearly distinguishing low- and high-temperature phases. In particular, we find $\lambda_L \sim T^{0.5}$ and $\lambda_L \sim T^{2.5-3}$ above and below the transitions.

2. The spatio-temporal evolution of the OTOC exhibits ballistic spreading of perturbation in the form of a linear light-cone throughout the temperature range for both easy-plane and easy-axis anisotropies, as shown in Figs.2(a),(b), above and below $T_{KT}$, for $\Delta < 1$. Unlike typical quantum systems [34, 42, 50], we do not find any signature of broadening of the ballistic propagation front of OTOC, even close to the phase transitions. However, we find that there is a *delay* $t_0$ in the onset of the light-cone. The time scale $t_0$ increases with decreasing temperature and seems to diverge for $T \to 0$, like $1/\lambda_L$.

3. We find the butterfly speed $\nu_B$ has a non-monotonic temperature dependence, showing a minimum at the transitions. This is the only sharp signature of the phase transition detectable via many-body chaos.

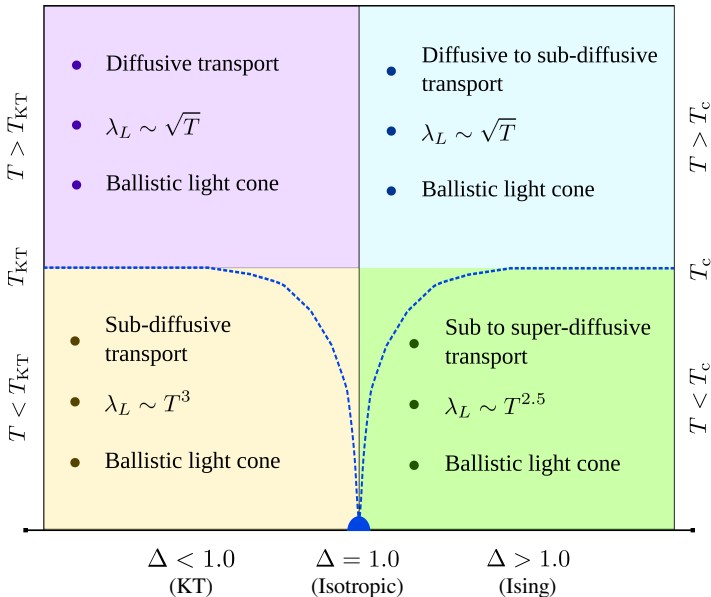

Figure 1: **Phase diagram:** Schematic phase diagram showing dynamical transitions and/or crossovers in terms of chaos and transport across KT and Ising transitions for easy-plane ($\Delta < 1$) and easy-axis ($\Delta > 1$) anisotropies.

4. Contrary to $\lambda_L(T)$, sharp signatures of the phase transitions is seen in $\tau(T)$, the time scale extracted from the temporal decay, $C_{xy}(t) = (1/N)\sum_{\mathbf{r}}\left(\langle S_{\mathbf{r}}^x(t)S_{\mathbf{r}}^x(0) + S_{\mathbf{r}}^y(t)S_{\mathbf{r}}^y(0)\rangle\right) \sim \exp(-t/\tau)$, above the transition for the auto-correlation function of the planar components of spins. This implies that the chaos time-scales $1/\lambda_L$, $t_0$ are unrelated to the relaxation time $\tau$. We find a power-law decay $C_{xy}(t) \sim 1/t^\alpha$ ($\alpha < 1$) for the easy-plane case below $T_{KT}$.

5. We show clear evidence of anomalous diffusion below and close to the transitions for the easy-axis case. We find sub-diffusive to super-diffusive ($\alpha > 1$) crossover across Ising transition $T_c$ for correlation function of the conserved out-of-plane component, $C_{zz}(t) = (1/N)\sum_{\mathbf{r}}\langle S_{\mathbf{r}}^z(t)S_{\mathbf{r}}^z(0)\rangle \sim 1/t^\alpha$. The correlation function $C_{zz}(t)$ shows oscillatory behaviour, expected from spin waves, below $T_{KT}$ for the easy-plane case. On the contrary, for both $\Delta > 1$ and $\Delta < 1$, $C_{zz}(t)$ always exhibits diffusive behaviour at high temperatures with $\alpha \approx d/2 = 1$, as expected for two dimensions ($d = 2$). We also corroborate the high-temperature diffusive behaviour by computing the dynamical correlation function $S^{zz}(\mathbf{q}, t) = \langle S_{\mathbf{q}}^z(t)S_{-\mathbf{q}}^z(0)\rangle$, where $S_{\mathbf{q}}^z(t)$ is Fourier transform of $z$-component of spins at time $t$. However, we find that the actual diffusion coefficient $D$ extracted from $S^{zz}(\mathbf{q}, t)$ is, in general, either quantitatively or qualitatively different from $\tilde{D} = v_B^2/4\lambda_L$. We find spin diffusion constant $D \simeq \tilde{D}$ only at infinite temperature for the easy-plane case in the XXZ model.

The above results indicate that there is no qualitative difference between KT and Ising transitions in terms of many-body chaos, at least for the range of anisotropies and temperature we access within our simulations. However, the dynamical spin-spin correlations show qualitatively very different behaviors in the KT and Ising ordered phases, within the time scale over which the perturbation spreads throughout the entire system for the system sizes studied. These imply that, relation between chaos and transport is much more intricate for phases with anomalous diffusion, unlike that in the high-temperature phase well above the transitions, where the diffusive behavior of spin correlation can be linked with the ballistic spread of chaos [14, 15].

## 4 Results

We study the model Eq.(1) with $J = 1$ and periodic boundary condition for two values of anisotropy, $\Delta = 0.3$ (easy plane) and 1.2 (easy axis), for square lattices with $N = L^2$ sites, with $L = 32, 64, 128$. We generate $10^4$ initial equilibrated configurations at each $T$ via Metropolis Monte Carlo (MC) simulations, and time evolve the configurations via Eq.(2) using fourth-order Runge-Kutta method with time step $\Delta t = 0.005$. As already mentioned, we look into two types of correlation functions – (1) The dynamical spin correlation functions, $C_{xy}(t)$, $C_{zz}(t)$, $S^{zz}(\mathbf{q}, t)$, and (2) The classical OTOC of Eq.(3).

### 4.1 Thermodynamics

We first characterize the thermodynamic phases from MC simulations. In particular, we estimate the KT and Ising transition temperatures $T_{KT} \simeq 0.74$ for $\Delta = 0.3$ and $T_c \simeq 0.96$ for $\Delta = 1.2$, respectively [see Appendix A]. We mainly focus close to the phase transitions and carry out the calculations below and above the transitions for a range of temperatures $0.5 \lesssim T \lesssim 2.0$, in the KT and Ising-ordered phases as well in the paramagnetic phase. In the easy-plane case ($\Delta = 0.3$), one expects the dynamics in the low-temperature phase to be controlled by gapless spin waves [58, 60] which lead to algebraic spatial correlation $\langle \mathbf{S_r}(0) \cdot \mathbf{S_0}(0)\rangle \sim r^{-\eta}$, where the exponent $\eta = T/(2\pi\rho_s)$ and $\rho_s$ the spin stiffness (see Appendix A). The KT transition occurs due to vortex-antivortex unbinding, resulting into a vortex plasma phase for $T \gtrsim T_{KT}$ [35, 61, 62], where the dynamics is expected to be dictated by the motion of free vortices. We obtain $T_{KT}$ from the universal Nelson-Kosterlitz jump crite-

rion [63] (Fig.9, Appendix A). The statics and dynamics are qualitatively different for easy-axis anisotropy $\Delta = 1.2$. We obtain the two-dimensional (2d) Ising transition temperature $T_c$ from divergence of specific heat and vanishing of the order parameter $m_z = (1/N)\sum_{\mathbf{r}}\langle S_{\mathbf{r}}^z \rangle$ (Appendix A). The spin waves of the Ising-ordered phase have a gap $\Delta_0 = 4(\Delta - 1)$ (Appendix A), and the spatial correlation decays exponentially with distance for all temperatures except at $T_c$. Below we investigate how these well-known static and dynamic properties of the model influence transport and chaos in the two cases.

## 4.2 Many-body chaos

We demonstrate the growth and the spread of initial perturbation at $\mathbf{r} = 0$ via $\mathcal{D}(\mathbf{r} = x\hat{\mathbf{x}}, t)$ for a 1d cut along $x$ direction at $T = 2.0 > T_{\mathrm{KT}}$ and $T = 0.5 < T_{\mathrm{KT}}$ in Fig.2 (a) and (b), respectively, for $\Delta = 0.3$. It is evident that, both below and above $T_{\mathrm{KT}}$, the chaos has a ballistic spread like a light cone. As evident from Fig.2(a) for $T = 2$, and as we have observed even for $T$ close to $T_{\mathrm{KT}}$ (not shown), the chaos front across the light cone remains sharply defined and we do not see any evidence for broadening of the front, unlike the diffusive broadening seen for quantum

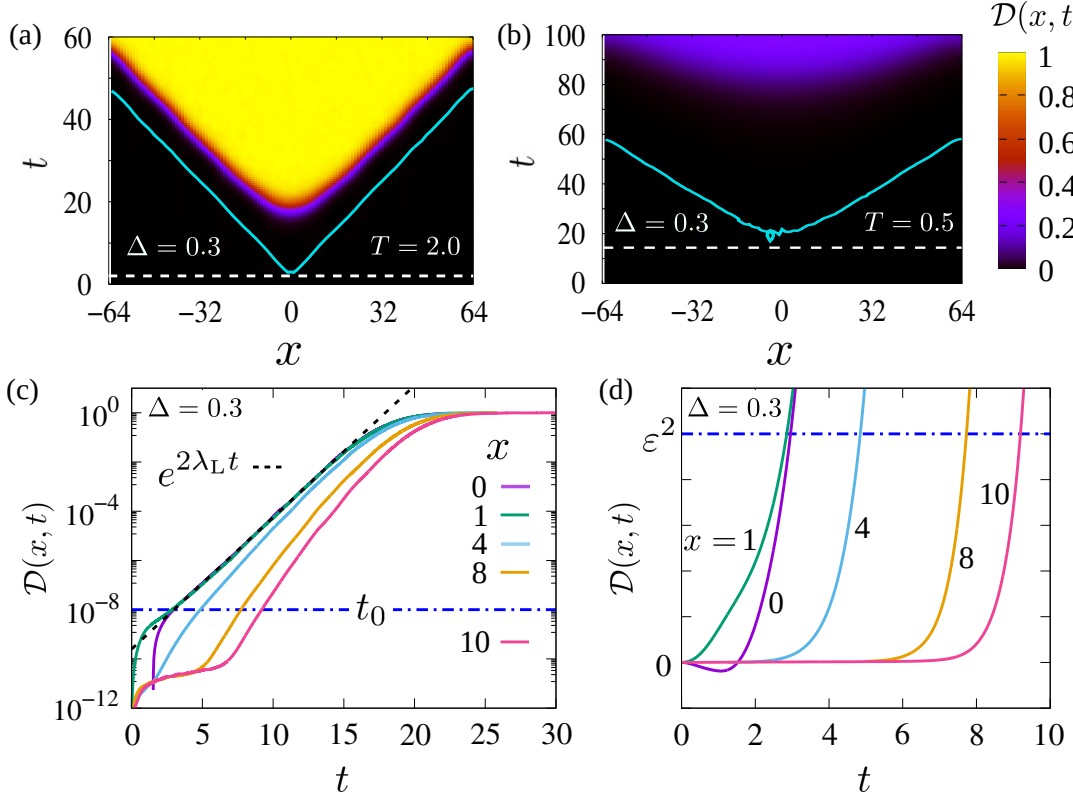

Figure 2: **Spatio-temporal evolution of classical OTOC in 2d XXZ model:** (a) and (b) show the growth and spread of initial perturbation at the origin for $\Delta = 0.3$ at temperature $T = 2.0$ $(> T_{\mathrm{KT}})$ and $T = 0.5$ $(< T_{\mathrm{KT}})$, respectively. The color denotes the value of classical OTOC, $\mathcal{D}(x, t)$, along a one dimensional (1d) cut in the $x$ direction for a perturbation strength $\varepsilon = 10^{-4}$. The solid lines are the light cones obtained from the generalized Lyapunov exponent for $\lambda_{\mathrm{L}}(x, t) = 0$. The horizontal dashed lines denote the delay time $t_0$ for the onset of exponential growth. (c) The time evolution of $\mathcal{D}(x, t)$ at $x = 0, 1, 4, 8, 10$. The dashed line is the exponential fit to obtain $\lambda_{\mathrm{L}}$ from $\mathcal{D}(0, t)$. The dashed-dotted line denotes the delay time $t_0$. (d) Zoomed-in view of $\mathcal{D}(x, t)$ at early times. $\mathcal{D}(0, t)$ initially becomes negative.

systems with short-range interactions and finite local Hilbert space [34, 42, 50]. We observe the same phenomena for easy-axis anisotropy $\Delta = 1.2$ as shown in Fig.10, Appendix B.

To get a better look at the spatio-temporal evolution of the perturbation, we plot $\mathcal{D}(x,t)$ as a function of $t$ for a few $x$ in Fig.2(c). Due to the choice of the orthogonal perturbation, $\mathcal{D}(0,t)$ starts from zero and initially becomes negative [Fig.2(d)] over an early-time regime, followed by a power-law growth (linearly with $t$, not demonstrated) till $t_0$, before it starts growing exponentially from a value $\mathcal{D}(0,t_0) \simeq \varepsilon^2$. As evident from Fig.2(c), the exponential growth ensues at a later time for $x \neq 0$.

**Lyapunov exponent**: To quantify spatio-temporal profile of chaos, we define a generalized Lyapunov exponent

$$\lambda_{\mathrm{L}}(x,t) = \frac{1}{2t} \ln\left[ \frac{\mathcal{D}(x,t)}{\varepsilon^2} \right]. \tag{4}$$

Using the above, we obtain a light cone from the locus of $\lambda_{\mathrm{L}}(x,t) = 0$, i.e. where the generalized Lyapunov exponent crosses zero or $\mathcal{D}(x,t) = \varepsilon^2$, as plotted in Figs.2(a),(b). At low temperature $T = 0.5$ [Fig.2(b)], the tip of the light cone at $x = 0$ gets rounded, and, more importantly, shifts to a later time $t_0$, compared to that at $T = 2.0$ [Fig.2(a)] (also see Fig.3(c)). This clearly suggests a temperature-dependent delay $t_0$ in the onset of the light cone. We also find similar time scale from $\langle(\delta \mathbf{S}_x(t))^2\rangle$ as shown in Fig.12, Appendix B. As mentioned in Sec.2, the quantity $\langle(\delta \mathbf{S}_x(t))^2\rangle$ starts from $\varepsilon^2$ at $x = 0$, $t = 0$. But, it initially decreases with time at $x = 0$, just like $\mathcal{D}(0,t)$ in Figs.2(c),(d).

We extract the Lyapunov exponent $\lambda_{\mathrm{L}}(T)$ as a function of temperature by fitting $\mathcal{D}(0,t) \sim \varepsilon^2 e^{2\lambda_{\mathrm{L}} t}$ in the exponential growth regime, e.g. in Fig.2(c). The results are shown in Figs. 3(a),(b) across the KT and Ising transitions, respectively, for different system sizes. A smooth crossover around the transitions can be clearly seen indicating a change of temperature dependence of $\lambda_{\mathrm{L}}$. We find $\lambda_{\mathrm{L}} \sim T^{2.86}$ for $T \leq T_{\mathrm{KT}}$ in the KT phase for $\Delta = 0.3$ and $\lambda_{\mathrm{L}} \sim T^{2.45}$ in the Ising ordered case for $\Delta = 1.2$. For the latter, the spin-wave spectrum has a gap $\Delta_0 \simeq 0.8$ (Appendix A), and we expect [41, 42] an activated $T$ dependence, $\lambda_L \sim e^{-\Delta_0/T}$, possibly with a power-law pre-factor, at low temperatures $T \ll \Delta_0$. However, for relatively high temperature $T \gtrsim 0.5$, studied here, we presumably capture only the power-law pre-factor $\sim T^{2.45}$ in Fig.3(b). In both the easy-axis and easy-plane cases, $\lambda_{\mathrm{L}}(T)$ is consistent with a $\sqrt{T}$ dependence above the transitions, albeit over a limited range of temperature. At very high temperature $T \gtrsim 2$, $\lambda_{\mathrm{L}}$ eventually saturates to the infinite temperature value $\sim 1$ [Fig.3(a),(b)], which we calculate separately by doing spin-dynamics simulations starting with completely random initial configurations $\{\mathbf{S}_{ar}(0)\}$. The $\sqrt{T}$ dependence for the Lyapunov exponent has also been seen for the classical spin liquid phase of a frustrated spin system [15]. The results in Figs. 3(a),(b) indicate no significant effect of $L$ and critical slowing down on $\lambda_{\mathrm{L}}$, unlike that observed across liquid-gas critical point [30]. However, the results imply that the individual phases can still be distinguished in terms of $\lambda_{\mathrm{L}}(T)$ [10–13]. Thus many-body chaos indeed could be an additional tool to characterize dynamics in phases in classical systems and may give new insights not contained within traditional static and dynamical properties. The crossover in chaos across KT transition has been studied earlier [19, 20, 22], either for smaller system sizes or with a different dynamics. Similar crossover in $\lambda_{\mathrm{L}}(T)$ has been reported for the Ising transition with various different types of dynamics [18, 23, 24, 29].

**Butterfly speed:** We next show the temperature dependence of the the butterfly speed $v_B(T)$ in Fig.3(d). The light cones, i.e. the locus of $\lambda_{\mathrm{L}}(x,t) = 0$, at a few temperatures, e.g., as shown in Fig.3(c), are fitted using $t = x/v_B + t_0$ with $v_B$ and $t_0$ as fitting parameters (see Figs.11(a),(b), Appendix B for more details). The speed $v_B$ exhibits a non-monotonic temperature dependence, having a *broad* minimum around the KT and Ising transition temperatures. A non-monotonic behavior in $v_B(T)$ has been observed in Ref. [18] for the finite-temperature 2d

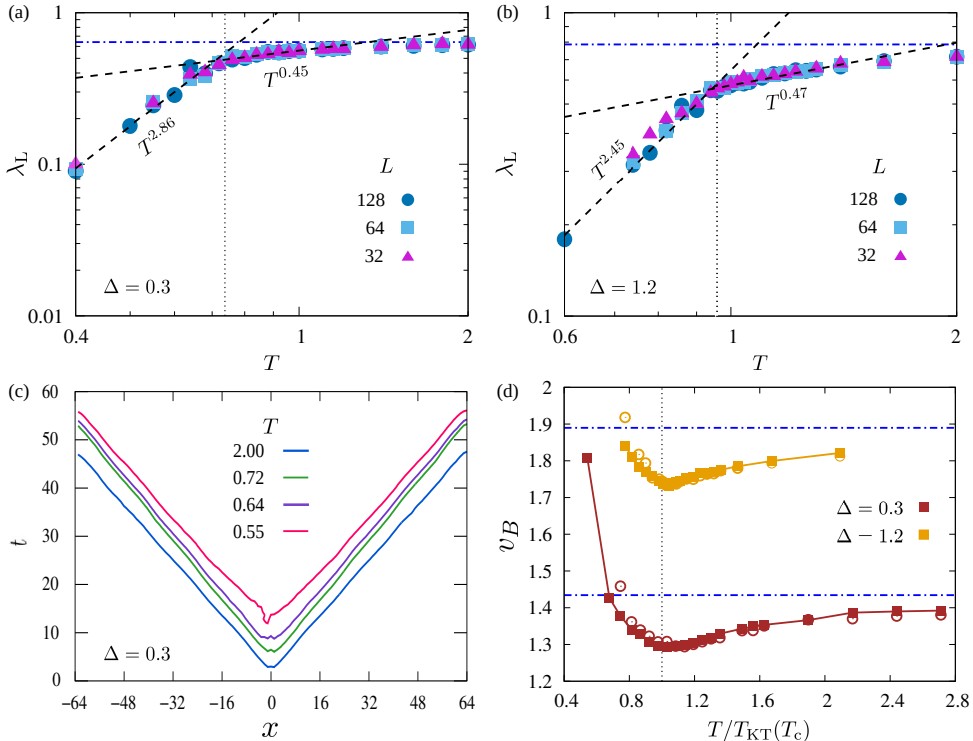

Figure 3: **Temperature dependence of the Lyapunov exponent and the butterfly speed:** (a) and (b) show the temperature dependence of $\lambda_{\mathrm{L}}$ across the KT and Ising transitions for $\Delta = 0.3$ and $\Delta = 1.2$, respectively. Results are shown for three different system size, $L = 128$ (circle), 64 (square) and 32 (triangle). Power law fits have been obtained for $T < T_{\mathrm{KT}}(T_c)$ and $T > T_{\mathrm{KT}}(T_c)$ (dashed lines). The dashed-dotted line represents the value of $\lambda_{\mathrm{L}}$ at infinite temperature. (c) Ballistic light cones ($\lambda_{\mathrm{L}}(x, t) = 0$) at different temperature across the KT transition. The butterfly speed, $v_{\mathrm{B}}$, and the delay time, $t_0$, are found from a linear fit to the light cones. (d) Temperature dependence of butterfly speed for easy-plane and easy-axis anisotropies across KT and Ising transitions, respectively, for $L = 128$ (square) and 64 (open circle). The dashed-dotted line denotes the value of $v_B$ at infinite temperature. Minima are observed at the transitions (vertical dotted line).

Ising transition in the classical limit of $O(1)$ model. However, in contrast to our results, there $v_B(T)$ shows a maximum at the transition for $O(1)$ model. This implies that chaotic properties are dependent on the details of the dynamics even close to critical points. As discussed in the Appendix D, one can obtain dynamical scaling laws for OTOC across finite temperature transitions with diverging length and time scales, as in the case of quantum phase transition [53]. Based on these scaling laws, or even just simple dimensional argument [41], $v_B \sim \xi/\xi^z = \xi^{1-z}$, with dynamical exponent $z \geq 1$. Similarly, for $\xi \gg L$, i.e. close to the transitions, $v_B \sim L^{1-z}$, giving the finite-size scaling of $v_B$. As mentioned earlier, $z = 1$ for the easy-plane case [58,59], thus the weak system size dependence of $v_B(T)$ ($\sim L^0$) in the KT phase ($T \leq T_{\mathrm{KT}}$) for $\Delta = 0.3$ in Fig.3(d) is consistent with the scaling law. However, the same features in $v_B(T)$ are seen around $T_c$ for the easy-axis case $\Delta = 1.2$ [Fig.3(d)] where one expects $z = 4 - \eta$ [35,38] and much stronger dependence of $v_B$ on $|T - T_c|$ and $L$. This discrepancy could be due to the fact that the easy-axis anisotropy $\Delta = 1.2$ studied here is not large enough to access true critical regime expected over a narrow range of $T$ around $T_c$. Also, dynamics for such a large $z \approx 4$ becomes extremely slow and thus it may be difficult to capture the asymptotic critical dynamics within our simulations times. We note that Ref. [18] also finds very weak temperature and

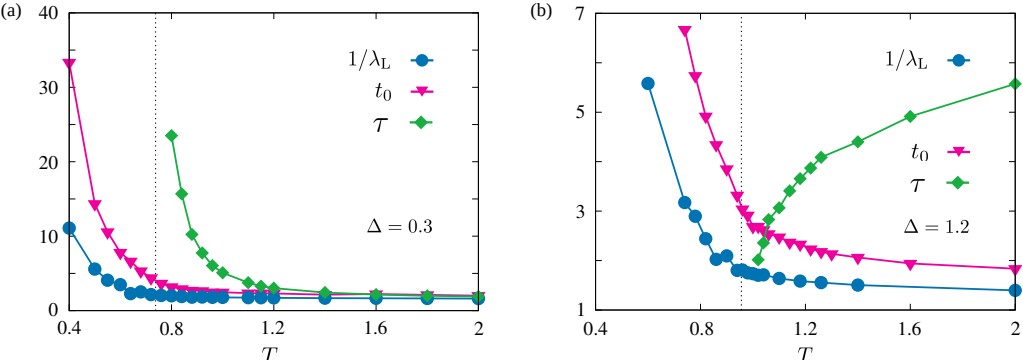

Figure 4: **Comparison of chaos and relaxation time scales:** The $T$ dependence of the delay time $t_0$, Lyapunov time $\lambda_L^{-1}$ and the relaxation time $\tau$, extracted from spin auto-correlation function $C_{xy}(t)$ above the transitions, are shown for (a) easy-plane ($\Delta = 0.3$) and (b) easy-axis ($\Delta = 1.2$) anisotropies. Vertical dotted lines denote the transitions.

system-size dependence for $v_B(T)$ close to $T_c$ in 2d $O(1)$ model, where $z = 2$ and stronger variations with $|T - T_c|$ and $L$ are expected for $v_B(T)$. We keep more detailed analysis of the scaling laws for OTOC and $v_B$ for future studies.

At high temperature we find that the quantity $\tilde{D} = v_B^2/4\lambda_L$ to be temperature independent, as discussed later [Fig.8], suggesting $v_B \sim T^{0.25}$, similar to that found in a classical spin liquid phase [15]. The $v_B(T)$ in Fig. 3(d) suggests faster spread of chaos at low temperatures. This could be due to well-defined spin-wave excitations in the low-temperature KT and Ising-ordered phases. In this regime $v_B$ increases at lower temperature whereas $\lambda_L \rightarrow 0$ as $T \rightarrow 0$, implying a large $v_B/\lambda_L \ell$ ($\ell \simeq 1$, the lattice spacing). This feature may persist even for quantum XXZ model with small $S$ and thus one maybe able to observe [47] the exponential growth in the quantum limit for such a regime dominated by weakly interacting spin waves.

The delay time $t_0$ extracted from the light cones [Fig.3(c)] is shown as a function of temperatures in Figs.4(a) and (b). The existence of the delay time and the linear form of the light cone for $t > t_0$ are further corroborated by plotting $\lambda_L(x, t)$ as a function of $x/(t - t_0)$ in Figs.5(a),(b). The $\lambda_L(x, t)$ for different $t$ collapses on a single curve near the light cone $\lambda_L(x, t) = 0$. We also find that, sans the region deep inside the light cone, $\mathcal{D}(x, t)$ can be fitted with a ballistic form $\varepsilon^2 \exp[\lambda_L t\{1 - (x/v_B(t - t_0))^\nu\}]$ for $t > t_0$ for both easy-plane and easy-axis cases, as discussed in the Appendix.C. The exponent $\nu$ changes from 2 to 1 going from high to low temperatures across the transitions, as shown in Figs.14(b),(c) in Appendix.C. Surprisingly, as shown in Fig.4(a) and (b), $t_0$, which characterizes early-time regime prior to exponential growth, roughly follows the temperature dependence of $\lambda_L^{-1}$, especially at high temperature. Naively, one would expect $t_0 \sim 1/J \sim 1$, a microscopic time scale. But, $t_0$ tends to diverge for $T \rightarrow 0$ [Figs.4(a),(b)]. Thus interaction effects, that lead to chaotic growth, presumably influence the pre-chaotic non-exponential growth regime of $\mathcal{D}(x, t)$ too.

We have also separately computed $\langle (\delta S_x^i(t))^2 \rangle = \langle (S_{ax}^i(t) - S_{bx}^i(t))^2 \rangle / 2$ for $i = x, y, z$. All the components give the same $\lambda_L(t)$ and $v_B(T)$ and exhibit qualitatively same behaviour, unlike the planar and out-of-plane auto-correlation functions $C_{xy}(t)$ and $C_{zz}(t)$ that we discuss below.

## 4.3 Dynamical spin-spin correlations, diffusion and anomalous diffusion

To understand the possible connection of growth and spread of chaos to transport and dynamical correlations, we first look into $C_{xy}(t)$ and $C_{zz}(t)$ at various temperatures, as shown

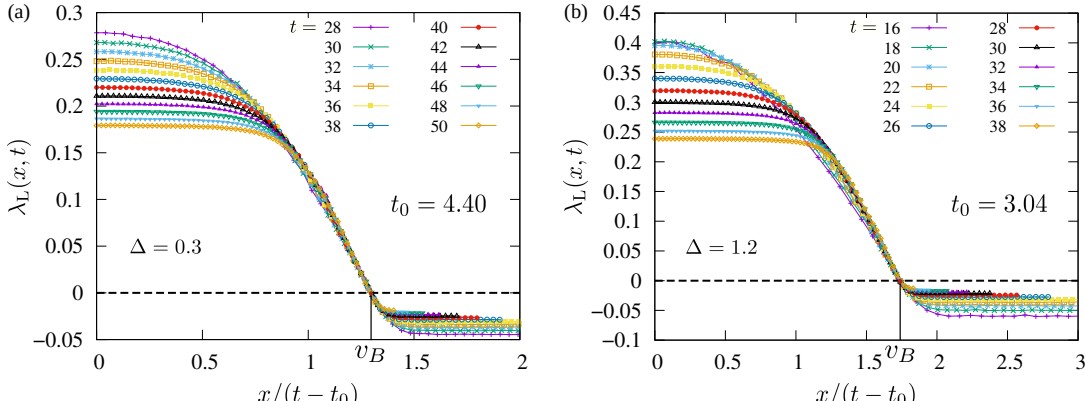

Figure 5: **Scaling collapse of generalized Lyapunov exponent:** $\lambda_{\mathrm{L}}(x,t)$ for different $t$ collapses into a single curve near the light cone ($\lambda_{\mathrm{L}}(x,t) = 0$) when plotted as function of $x/(t - t_0)$. $t_0$ has been extracted from the linear fits to the light cones, e.g. in Fig.11(b), Appendix B. (a) $\Delta = 0.3$, $T = 0.72$ and (b) $\Delta = 1.2$, , $T = 0.96$. The collapse corroborates the existence of linear light cone with a onset time $t_0$.

in Fig.6(a) for $\Delta = 0.3$. For $T > T_{\mathrm{KT}}$, $C_{zz}(t) \sim 1/t^\alpha$, i.e. $C_{zz}(t)$ exhibits a power-law decay at long times ($t \gtrsim 10$). The exponent $\alpha \approx 1$ [Fig.6(b)] is consistent with the expected diffusive behaviour for $C_{zz}(t)$. Above the KT transition, $C_{xy}(t)$ decays exponentially with a time scale $\tau$ [see Fig.15 (a), Appendix E]. The evidence of critical slowing down can be observed in $\tau(T)$, as shown in Fig.4(a). Since $C_{xy}(t)$ approaches a finite value $C_{xy}(t \to \infty) = C_{xy}^\infty$ (see Appendix E) in the long time limit below the transitions, we plot $\widetilde{C}_{xy}(t) = C_{xy}(t) - C_{xy}^\infty$ in Fig.6(a) for $T < T_{\mathrm{KT}}$. $C_{xy}^\infty \neq 0$ for $T < T_{\mathrm{KT}}$ due to strong finite-size effect [64]. $\tilde{C}_{xy}(t)$ shows a power law decay at long times with $\alpha < 1$, as shown in Fig.6(b). However, we could extract $\alpha$ only close to $T_{\mathrm{KT}}$ due to large error in estimating $C_{xy}^\infty$ for $T \lesssim 0.5$. Qualitatively, the power-law decay of $C_{xy}(t)$ is expected from non-interacting gapless spin waves in the KT phase giving rise to a temporal spin correlation $C_{xy}(t) \sim t^{-\eta}$ [60], implying $\alpha \simeq \eta = T/(2\pi\rho_s)$. However, our results for $\alpha$ close to $T_{\mathrm{KT}}$ does not quantitatively match the non-interacting spin-wave results for $\eta(T)$ plotted in Fig.6(b). This could be due to coupling between planar and out-of-plane components via spin-wave interactions or finite time ($\lesssim 100$) accessed in our simulations. The asymptotic power law may set in at very long times, as well known for Heisenberg chain [65]. To verify whether a relatively steady power-law regime is reached for $\widetilde{C}_{xy}(t)$, we also look into a time-dependent exponent or local logarithmic slope $\alpha(t) = d\ln(\widetilde{C}_{xy}(t))/d\ln(t)$ [Fig.16(a), Appendix E]. $\alpha(t)$ has a clear drift towards a larger value. This implies that, below $T_{\mathrm{KT}}$, long-time asymptote for $C_{xy}(t)$ is not reached for the time scales over which the chaos spreads ballistically through our finite-sized systems. We note that the transient power-law regime [$20 \lesssim t \lesssim 80$, Fig.16(a), Appendix E], though partially overlaps with the non-exponential growth regime $t < t_0 \sim 5 - 35$ [Fig.4(a)], but extends far beyond the latter and continues deep inside the light cone. Thus the observed power-law ($\alpha < 1$) regime persists over the time window of the ballistic spread of chaos over the system sizes considered here. We could not extract any power-law exponent for $C_{zz}(t)$ below $T_{\mathrm{KT}}$ since it becomes small and oscillatory at low temperatures (not shown).

We did similar calculations of $C_{xy}(t)$ and $C_{zz}(t)$ for the easy-axis case ($\Delta = 1.2$). As in the easy-plane case, $C_{zz}(t)$ [Fig.15(b), Appendix E] shows a diffusive power law with $\alpha \simeq 1$ at high temperatures ($T \gg T_c$), as shown in Fig.6(b). However, $\alpha$ decreases approaching the transition indicating a sub-diffusive behaviour. In this regime, the local slope $\alpha(t)$ only shows slight drift with $t$ as shown in Fig.16(b), Appendix E. Below $T_c$, again we obtain $\tilde{C}_{zz}(t) = C_{zz}(t) - C_{zz}^\infty$

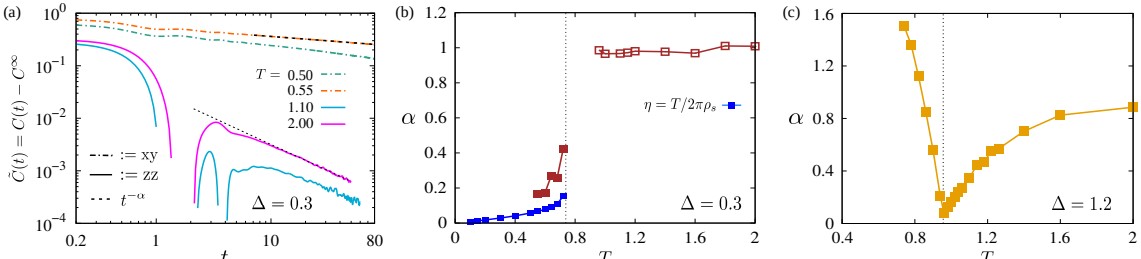

Figure 6: **Spin auto-correlation function and anomalous diffusion:** (a) Spin-spin auto-correlation function across KT transition for $\Delta = 0.3$ and $L = 128$. At long times ($t \gtrsim 10$) the auto-correlation exhibits a power-law decay, $\tilde{C}(t) \sim 1/t^{\alpha}$, as indicated by the black dashed line for $T = 2.0$ and $0.50$. (b) The temperature dependence of $\alpha$ across the transitions. For the easy-plane anisotropy, $\alpha \approx 1$ is extracted from $C_{zz}(t)$ above $T_{KT}$. Below $T_{KT}$, $\alpha < 1$ has been extracted from $\tilde{C}_{xy}(t)$ and $\alpha < 1$. For $T < T_{KT}$, $\alpha$ is compared with the exponent $\eta = T/2\pi\rho_s$, expected from non-interacting gapless spin waves. For the easy-axis case (c), $\alpha$ has diffusive $\rightarrow$ sub-diffusive $\rightarrow$ super-diffusive ($\alpha > 1$) crossovers from high to low temperature across $T_c$. In this case, $\alpha$ is extracted from $\tilde{C}_{zz}(t)$. The vertical dotted lines mark the transitions.

[Fig.15(b), Appendix E] by subtracting the $t \rightarrow \infty$ value of $C_{zz}(t)$. In the Ising case, $C_{zz}^{\infty} \neq 0$ below $T_c$ because of the symmetry breaking. We find a surprising crossover from sub to super-diffusive scaling of $\widetilde{C}_{zz}(t)$, with $\alpha(T)$ increasing rather sharply from $\alpha \ll 1$ to a value greater than one below $T_c$, as shown in Fig.6(c). The analysis of $\alpha(t)$ [Fig.16(b), Appendix E] indicates a steady exponent corroborating the super-diffusive power law. The latter may again be an intermediate-time behavior, but it happens over the same times scale over which the chaos spreads in the system. We do not have any good understanding of the anomalous sub and super-diffusive behaviour across the transition and in the Ising ordered phase. For the latter, the phenomena may be arising from some additional conserved mode emerging in the ordered phase and nonlinear coupling between the hydrodynamic modes as happens in one dimensional XXZ model [66]. The planar correlation $C_{xy}(t)$ decays exponentially with the decay time $\tau(T)$ [Fig.4(b)] for $T > T_c$. However, in contrast to the easy-plane case [Fig.4(a)], $\tau(T)$ sharply decreases approaching the Ising transition. We note that the planar components are not the critical modes for $\Delta > 1$ and hence the associated relaxation time does not necessarily need to show critical slowing down. $C_{xy}(t)$ has strongly oscillatory behaviour for $T < T_c$ (not shown).

Finally, to probe further the high-temperature diffusive phase and the relation between $\tilde{D} = v_B^2/4\lambda_L$, which superficially looks like a diffusion constant from dimensional ground, and the actual spin diffusion coefficient $D$, we compute the dynamical structure factor (or its Fourier transform)

$$S^{zz}(\mathbf{q}, t) = \frac{1}{N} \sum_{\mathbf{r}, \mathbf{r}'} e^{i\mathbf{q}\cdot(\mathbf{r}-\mathbf{r}')} \langle S_{\mathbf{r}}^z(t) S_{\mathbf{r}'}^z(0) \rangle. \tag{5}$$

For computing the above from spin dynamics simulation, we rewrite above expression as $S^{zz}(\mathbf{q}, t) = \langle S_{\mathbf{q}}^z(t) S_{-\mathbf{q}}^z(0) \rangle$. Here $S_{\mathbf{q}}^z(t)$ is Fourier transform of $z$-component of spins at time $t$. We obtain $S_{\mathbf{q}}^z(t)$ from a configuration $\{S_{\mathbf{r}}^z(t)\}$ at time $t$, and average over configurations to obtain $S^{zz}(\mathbf{q}, t)$ in Eq.(5). If the system exhibits normal diffusion, for large wavelength $\mathbf{q} \rightarrow 0$ we expect $S^{zz}(\mathbf{q}, t)$ to decay exponentially in time, i.e., $e^{-\kappa(\mathbf{q})t}$ where the decay rate $\kappa(\mathbf{q}) = Dq^2$. We choose momenta $q = q_x$ along $x$ direction, close to $\mathbf{q} = 0$ and fit $S^{zz}(q, t)$ with the exponential form to get $\kappa(q)$ for a given $q$. Fig.7(a) shows $S^{zz}(q, t)$ as a function of $q$ for several values of $t$ in the easy-plane ($\Delta = 0.3$) case at $T = 2.0$. The Gaussian form, $S^{zz}(q, t) \sim e^{-Dq^2 t}$ is evident, implying diffusive behaviour. The exponential decay of $S^{zz}(q, t)$

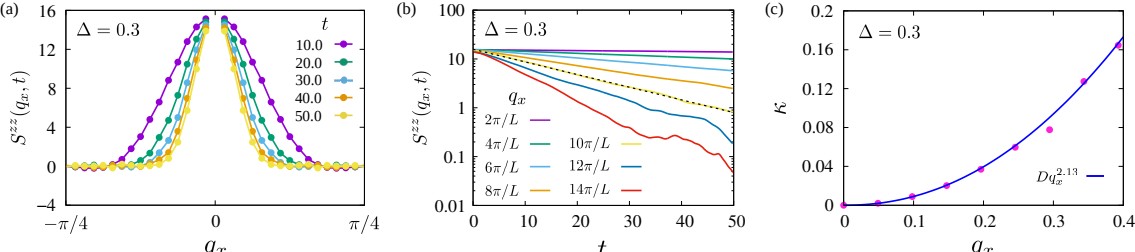

Figure 7: **Dynamical structure factor:** (a) Momentum dependence of $S^{zz}(q_x, t)$ at different times $t = 10$ to $50$. The behaviour is expected to be a Gaussian in $\mathbf{q}$ with standard deviation, $\sigma_{\mathbf{q}} = 1/\sqrt{2Dt}$ that decreases with time. (b) Exponential decay of $S^{zz}(q_x, t)$ in time for different momenta $q_x(k) = 2\pi k/L$, $k = 1, 2, \ldots, 7$ near $\mathbf{q} = 0$ along $x$ direction. We fit the data with $e^{-\kappa(q_x)t}$ (e.g. the dashed line) to extract $\kappa(q_x)$ (c) Quadratic dependence of $\kappa$ with $q_x$ near $\mathbf{q} = 0$. The fit with $\kappa = Dq_x^a$ gives $D = 1.22$ and $a = 2.13$. For the all the panels, we have taken $T = 2.00$ and $\Delta = 0.3$.

as a function of $t$ for small momenta is shown in Fig.7(b) for the same parameter values. We extract the diffusion constant $D$ for a range of temperatures where $\kappa(q)$ could be fitted via $Dq^a$ with $a \approx 2$. As we show in Fig.8(b) by plotting $a(T)$, $\kappa(q)$ follows the quadratic dependence in the high temperature regime, for both easy-plane and easy-axis cases, where the auto-correlation exponent $\alpha \simeq 1$ [Figs.6(b),(c)]. An example of near-quadratic dependence of $\kappa(q)$ is shown for $\Delta = 0.3$ and $T = 2.0$ in Fig.7(c). At lower temperatures, we find $\kappa(q)$ to deviate from the diffusive form and $S^{zz}(q, t)$ to exhibit oscillatory behaviour as a function of both $q$ and $t$ (not shown). The oscillatory behaviour is expected [58] below the transition and even slightly above it, due to spin-waves.

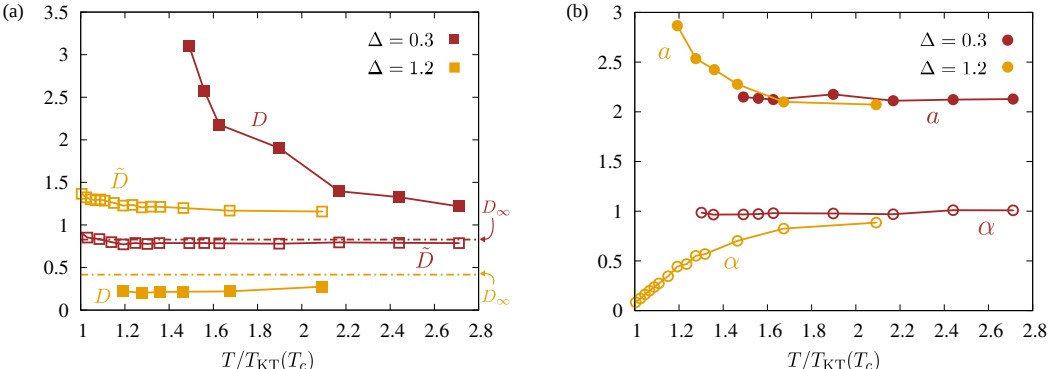

Figure 8: **Temperature dependence of diffusion constant:** The diffusion coefficient is extracted from the fit $\kappa(q) = Dq^a$ over a range at high temperatures. We plot in (a) $D$ (solid squares), and in (b) the exponent $a$ (solid circles), as a function of temperature. In (a), we show the comparison of diffusion constant $D(T)$ calculated from dynamical structure factor with $\tilde{D} = v_B^2/4\lambda_L$ (open squares) extracted from Fig.3 for $\Delta = 0.3$ and $\Delta = 1.2$. Horizontal dashed-dot lines represent the asymptotic infinite temperature values of $D_\infty$ at these two anisotropies. $D_\infty \simeq \tilde{D}_\infty$ ($\approx 0.8$, not shown) for easy-plane case whereas $D_\infty \approx 0.42$ and $\tilde{D}_\infty \approx 1.14$ (not shown) for easy-axis case. (b) shows that the deviation of the exponent $a$ from the diffusive value $\sim 2$ starts exactly where the auto-correlation exponent $\alpha$ (open circles) deviates away from $\sim 1$ (Fig.6). Note that for simplicity of notation, we refer to $D$ as diffusion constant even when $a$ deviates substantially from the diffusive value 2.

The diffusion constant $D$ calculated for both easy-plane and easy-axis anisotropies in the high-temperature diffusive regime ($\alpha \simeq 1$) is plotted in Fig.8, and compared with $\tilde{D}$, extracted from Fig.3. In the easy-axis case, both $D$ and $\tilde{D}$ are independent of temperature at high temperatures ($1 \lesssim T/T_c \lesssim 2$) and closely approach their infinite-temperature values, however, $D \ll \tilde{D}$. In the easy-plane case, $D$ varies substantially with temperature even at high temperature and slowly approaches its infinite-temperature value $D_\infty$. This behaviour is unlike that of $\tilde{D}$, which varies little with temperature and coincides with its infinite-temperature value $\tilde{D}_\infty$. Moreover, we find $\tilde{D}_\infty \simeq D_\infty$ for the easy-plane case. Nevertheless, our results suggest that $\tilde{D}$ is quite distinct from the actual diffusion constant $D$ in general, unlike that in a correlated classical spin-liquid state [15] or in quantum systems like strongly or weakly interacting diffusive metals [7,39,43,44]. For the strongly interacting metals, $\tilde{D}$ is related to the energy diffusion constant implying that the chaos or scarambling directly controls the thermal diffusion [7,39]. To this end, our results raise interesting questions about actual physical process governing $\tilde{D}$ in the semiclassical limit of 2d XXZ model. For example, it would be interesting to compute the energy diffusion constant for the spin model in future and see whether it corresponds more closely to $\tilde{D}$, rather than the spin diffusion constant that we calculate here.

## 5   Conclusion

We have studied here the OTOC, and the dynamical spin correlations in the semiclassical limit of the 2d XXZ model. In particular, we have tuned the anisotropy in the model to study the dynamical properties across KT and 2d Ising transitions via simulation of spin precession dynamics as a function of temperature. Thus we obtain a dynamical phase diagram in terms of chaos and spatio-temporal spin correlations for the classical 2d XXZ model. We have computed temperature dependence of the Lyapunov exponent and butterfly speed, which show crossover across the transitions and no effect of critical slowing down. Only relatively sharp signature of the transitions is exhibited by a non-monotonic temperature dependence of butterfly speed having a minimum at the transition.

Overall, we find chaotic growth and spread, and the dynamical correlations above the transitions at high temperature in the easy-plane and easy-axes cases very similar. However, the dynamical spin-spin correlations are very different at intermediate times in the low-temperature KT and Ising ordered phases, and close to the transitions, although chaos still spreads ballistically in these regimes. This leads us back to the question on the connection of chaos and transport. A simple, albeit heuristic, way [67] to obtain ballistic light-cone for chaos from diffusive transport is to take the *separable* ansatz $\mathcal{D}(x,t) \approx \varepsilon^2 e^{\lambda_L t} C_{zz}(x,t)$ for the OTOC and plug in the diffusive form $C_{zz}(x,t) \sim e^{-x^2/4Dt}/t^{d/2}$. This leads to a velocity-dependent generalized Lyapunov exponent [14,34,36] $\lambda_L(x,t) = \lambda_L\left(1 - (x/v_B t)^2\right)$, and naturally gives rise to the relation $D \sim \tilde{D} = v_B^2/4\lambda_L$. However, we find $\tilde{D}$ to be very different from $D$, except for the easy-plane case at infinite temperature. More importantly, the simple ansatz clearly fails in the phases exhibiting anomalous diffusion, like Ising ordered phases in the $XXZ$ model. Our observation of the anomalous diffusion over a large range of temperature for easy-axis anisotropy in the semiclassical limit of 2d XXZ model is intriguing and it would be good to get a proper understanding of these phenomena and their possible connections to chaos.

We reveal an early-time pre-exponential regime in the form of a temperature-dependent overall delay $t_0(T)$ in the onset of the light cone. The time scale $t_0$ is presumably connected with chaos time scale and originates from the same many-body interactions that give rise to chaotic growth. This result suggests the possibility of extracting new dynamical regime with a suitable choice of the correlation function. Such a regime may even exist for *non-chaotic* systems, e.g. integrable or *fully quantum* ones, where there is no exponential growth inside

the light cone [34].

In the absence of any good analytical understanding of many-body chaos in classical systems, our results call for the development of a theoretical framework [68] to compute the classical OTOC along the line of that done for quantum systems, in a suitable large-$N$ limit [16], or in some perturbative regime [36,68], like at low temperatures with weakly interacting spin waves. Such a theory may give rise to new insights into the dynamics of interacting classical systems as well as quantum systems in the semiclassical limit. It would be desirable to obtain hydrodynamic description of the OTOC in such classical spin systems with Hamiltonian dynamics or some related tractable toy models, e.g. with random classical Liouvillian dynamics, along the line of those developed for quantum systems [43, 48–51].

# Acknowledgements

We thank Subhro Bhattacharjee, Samriddhi Sankar Ray, Anupam Kundu, Sumiran Pujari and Sriram Ramaswamy for many useful discussions. We specially thank Anupam Kundu for critical reading of the manuscript and useful feedback. SB acknowledges support from The Infosys Foundation, India and SERB (ECR/2018/001742), DST, India.

**Funding information** SB acknowledges support from SERB (ECR/2018/001742), DST, India.

# A  Thermodynamic Properties

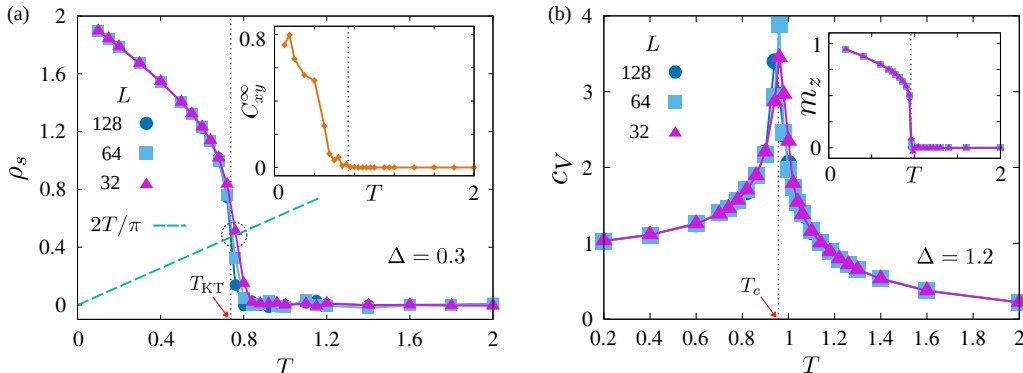

Figure 9: **Kosterlitz-Thouless (KT) and Ising transitions:** (a) The temperature dependence of spin stiffness $\rho_s$ for $\Delta = 0.3$. The KT transition temperature $T_{\mathrm{KT}}$ is obtained from the intersection between $\rho_s(T)$ and $2T/\pi$. We estimate $T_{\mathrm{KT}}$ from the largest system size ($L = 128$). Inset shows $C_{xy}^\infty$, the $t \to \infty$ value of $C_{xy}(t)$, as a function of temperature. (b) The specific heat $c_V$ as function of $T$ for $\Delta = 1.2$. The Ising transition temperature $T_c$ is estimated from the divergence of $c_V$, as well as from the magnetization $m_z$ (inset). The vertical dotted lines denote the transition temperatures.

## A.1 Spin stiffness and KT transition

To obtain the Kosterlitz-Thouless (KT) transition temperature $T_{KT}$ for $\Delta = 0.3$, we calculate the spin stiffness $\rho_s$, which measures the rigidity of the spin configuration to small twist or rotation of the spins along some direction. For the KT transition the relevant spin stiffness is obtained by twisting the planar components $(S^x_{\mathbf{r}}, S^y_{\mathbf{r}}) = (S^{\parallel}_{\mathbf{r}} \cos\phi_{\mathbf{r}}, S^{\parallel}_{\mathbf{r}} \sin\phi_{\mathbf{r}})$ of the spins. The spin stiffness for the model of Eq.1 in the main text can be obtained as

$$\rho_s(T) = \frac{J}{2N} \sum_{\mathbf{r},\boldsymbol{\delta}} \left\langle S^{\parallel}_{\mathbf{r}} S^{\parallel}_{\mathbf{r}+\boldsymbol{\delta}} \cos(\phi_{\mathbf{r}} - \phi_{\mathbf{r}+\boldsymbol{\delta}}) \right\rangle - \frac{J^2}{2NT} \sum_{\boldsymbol{\delta}} \left\langle \left( \sum_{\mathbf{r}} S^{\parallel}_{\mathbf{r}} S^{\parallel}_{\mathbf{r}+\boldsymbol{\delta}} \sin(\phi_{\mathbf{r}} - \phi_{\mathbf{r}+\boldsymbol{\delta}}) \right)^2 \right\rangle. \quad (6)$$

We calculate $\rho_s(T)$ via MC simulations and apply the Nelson-Kosterlitz universal jump criterion [63] $\rho_s(T_{KT})/T_{KT} = 2/\pi$ to obtain $T_{KT}$, as shown in Fig. 9(a).

## A.2 Two dimensional (2d) Ising transition

To estimate the 2d Ising transition temperature $T_c$ for $\Delta = 1.2$, we calculate the magnetization and the specific heat per site

$$c_V = \frac{1}{NT^2} \left( \langle \mathcal{H}^2 \rangle - \langle \mathcal{H} \rangle^2 \right), \quad (7a)$$

$$m_z = \frac{1}{N} \sum_{\mathbf{r}} \langle S^z_{\mathbf{r}} \rangle. \quad (7b)$$

$T_c$ is obtained from the divergence of $c_V$ shown in Fig. 9(b), as well as, from $m_z(T)$ which continuously goes to zero at $T_c$ [Fig. 9(b)(inset)].

## A.3 Spin-wave dispersion

**Easy-plane anisotropy**

In this case, the spin-wave dispersion is obtained by expanding the dynamical equations [Eq.2, main text] around the $T = 0$ ground state, which corresponds to all spins aligned along a direction (say $\hat{\mathbf{x}}$) in the $xy$-plane. In this case, $S^x_{\mathbf{r}} \simeq 1 \gg S^y_{\mathbf{r}}, S^z_{\mathbf{r}}$, and Eq.2 (main text) gets reduced to

$$\frac{dS^y_{\mathbf{r}}}{dt} = J \sum_{\boldsymbol{\delta}} (S^z_{\mathbf{r}} - \Delta S^z_{\mathbf{r}+\boldsymbol{\delta}}),$$

$$\frac{dS^z_{\mathbf{r}}}{dt} = J \sum_{\boldsymbol{\delta}} (S^y_{\mathbf{r}+\boldsymbol{\delta}} - S^y_{\mathbf{r}}),$$

Using Fourier transformation $S^i_{\mathbf{r}}(t) \rightarrow S^i_{\mathbf{q}}(\omega)$, it is straightforward to obtain the spin-wave dispersion for $q \rightarrow 0$ for the square lattice

$$\omega(\mathbf{q}) = 2J\sqrt{1-\Delta}q. \quad (8)$$

Hence the spin waves are gapless in the easy-plane case.

**Easy-axis anisotropy**

In this case, the all the spins can be taken to be aligned along $\hat{\mathbf{z}}$ direction, and $S^z_{\mathbf{r}} \simeq 1 \gg S^x_{\mathbf{r}}, S^y_{\mathbf{r}}$. Following the same method as the easy-plane case, we obtain the spin-wave dispersion

$$\omega(\mathbf{q}) = \Delta_0 + Jq^2, \quad (9)$$

with a spin-wave gap $\Delta_0 = 4J(\Delta - 1)$.

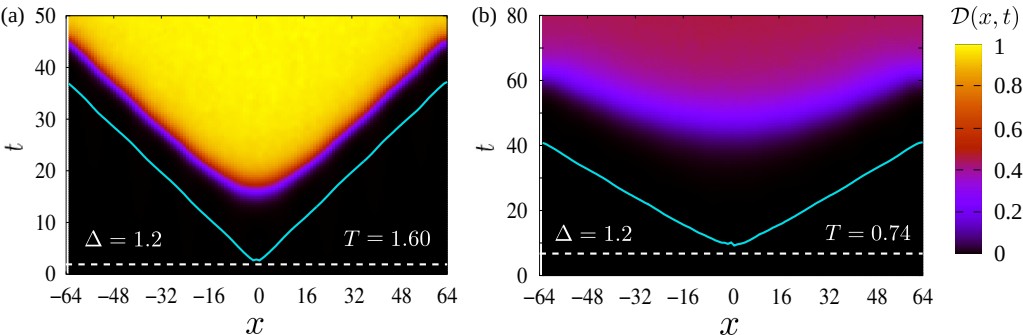

Figure 10: **OTOC and ballistic light cone for easy-axis anistropy:** $\mathcal{D}(x,t)$ at (a) $T = 1.60$ ($T > T_c$) and (b) $T = 0.74$ ($T < T_c$), for $\Delta = 1.2$. The solid lines are the light cones from extracted from $\lambda_{\mathrm{L}}(x,t) = 0$, and the horizontal dashed lines denote the delay time $t_0$.

## B    Classical OTOC

We show the OTOC $\mathcal{D}(x,t)$ for a cut along $x$ direction for $\Delta = 1.2$ at two temperatures, $T = 1.6 > T_c$ and $T = 0.74 < T_c$ in Fig.10(a),(b). The results are qualitatively similar to the easy-plane anisotropy $\Delta = 0.3$ [Fig.2(a),(b)] in the main text, namely the chaos spreads ballistically, as indicated by the light cones $\lambda_{\mathrm{L}}(x,t) = 0$, and the light cones start at a finite time $t_0$, which increases with decreasing temperature.

We extract the light cones, e.g. in Fig.3(c) (main text), by finding the locus $t_{\mathcal{D}_0}(x)$ of $\lambda_{\mathrm{L}}(x,t) = 0$ or $\mathcal{D}(x,t) = \mathcal{D}_0 = \varepsilon^2$, as shown in Fig.11(b) for $\Delta = 0.3$. The delay time $t_0$ and the butterfly speed $v_B$ are calculated by fitting the light cones in Fig.11(b) with the linear form $t_{\mathcal{D}_0} = x/v_B + t_0$. The results for $t_0$ and $v_B$ have been shown in the main text in Figs.4(a),(b) and Fig.3(d), respectively.

### B.1    Trajectory divergence and decorrelation

In all our calculations and results shown in the main text, we consider the decorrelation function [Eq.(3)] or the classical OTOC, which essentially measures how uncorrelated two spin configurations $a$ and $b$ are. As discussed earlier, the copy $b$ differs from $a$ initially only at a single site ($\mathbf{r} = \mathbf{0}$), namely $\mathbf{S}_{b\mathbf{r}}(0) = \mathbf{S}_{a\mathbf{r}}(0) + \varepsilon(\hat{\mathbf{n}} \times \mathbf{S}_{a\mathbf{0}})\delta_{\mathbf{r},\mathbf{0}}$. One can also consider the trajectory divergence, which is defined $\langle(\delta\mathbf{S}_{\mathbf{r}}(t))^2\rangle = \langle(\mathbf{S}_{b\mathbf{r}}(t) - \mathbf{S}_{a\mathbf{r}}(t))^2\rangle$. The decorrelation function and trajectory divergence differ at $\mathcal{O}(\varepsilon^2)$ initially at $\mathbf{r} = \mathbf{0}$, i.e.

$$\langle(\delta\mathbf{S}_{\mathbf{r}}(0))^2\rangle = 2\mathcal{D}(\mathbf{r},0) + \varepsilon^2\delta_{\mathbf{r},\mathbf{0}}. \tag{10}$$

This initial-time difference eventually relaxes in the intermediate time regime where chaos sets in and we get the same exponential growth and ballistic spread characterized by $\lambda_{\mathrm{L}}$ and $v_B$, respectively, from both of these quantities.

To compare the decorrelation function and the trajectory divergence, we plot $\langle(\delta\mathbf{S}_x(t))^2\rangle$ in Fig.12 as a function of $t$ for a few $x$, for $T = 2.00, 0.50$ and $\Delta = 0.3$. $\langle(\delta\mathbf{S}_x(t))^2\rangle$ exhibits behaviour very similar to $\mathcal{D}(x,t)$ [Fig.2(c),(d)]. By definition, $\langle(\delta\mathbf{S}_0(t))^2\rangle$ starts from $\varepsilon^2$ and decreases over an early-time regime, followed by a power-law growth till $t_0$, before it starts growing exponentially from a value $\mathcal{D}(0,t_0) \simeq \varepsilon^2$. The exponential growth occurs at a later time for $x \neq 0$. As shown in Fig. 13(a),(b) for $T = 2$ and $\Delta = 0.3$, both the decorrelation function and trajectory divergence give same values of $\lambda_{\mathrm{L}}$ and $v_B$.

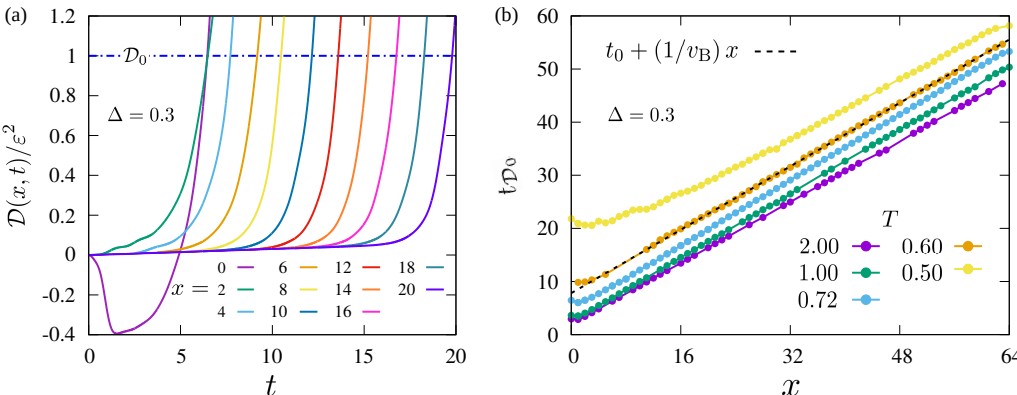

Figure 11: **Extraction of the butterfly speed and the delay time:** (a) shows the time dependence of $\mathcal{D}(x,t)/\varepsilon^2$ at different sites ($x = 0, 2, 4, \dots$) along a 1d cut in the $x$-direction at $T = 0.72$ for easy-plane anisotropy. Due to the choice of our orthogonal perturbation, $\delta \mathbf{S}_0 = \varepsilon(\hat{\mathbf{n}} \times \mathbf{S}_0)$, $\mathcal{D}(x,t)$ starts from 0 and increases to reach $\mathcal{D}_0 = \varepsilon^2$ in time $t_{\mathcal{D}_0}$. Initially $\mathcal{D}(x,t)$ becomes negative, before rising sharply to $\mathcal{D}_0$. (b) shows the light cones $t_{\mathcal{D}_0}$ obtained from the locus of $\lambda_{\mathrm{L}}(x,t) = 0$ or $\mathcal{D}(x,t) = \mathcal{D}_0$. The light cones are fitted with $t_{\mathcal{D}_0} = t_0 + x/v_B$ to extract $v_B(T)$ and $t_0(T)$. We get similar behaviour for easy-axis case.

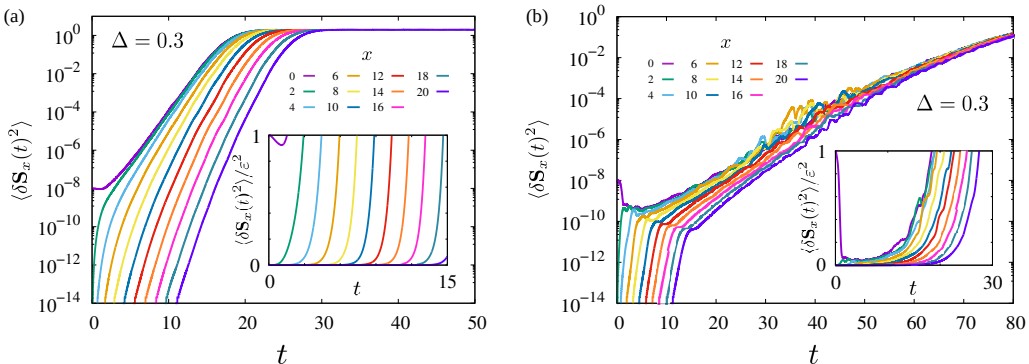

Figure 12: **Trajectory divergence:** We plot the time evolution of conventional trajectory divergence, $\langle (\delta \mathbf{S}_x(t))^2 \rangle$ at different sites $x = 0, 2, 4, \dots, 20$ for (a) $T = 2.00$ and (b) $T = 0.50$ for easy-plane anisotropy $\Delta = 0.3$. In the insets, we show zoomed-in view of the early time evolution of the same quantity.

## C  Velocity-dependent Lyapunov exponent

As shown in Fig.5(a),(b), the generalized Lyapunov exponent $\lambda_{\mathrm{L}}(x,t)$ [Eq.(4)] at different $t$ can be collapsed into a single curve as a function of a velocity $v = x/(t - t_0)$ for $t > t_0$ over a relatively large range around $v = v_B$ for both outside ($v > v_B$) and inside ($v < v_B$) the light cone. However, this range shrinks progressively with decreasing temperature. Over this range, the velocity-dependent Lyapunov exponent $\lambda_{\mathrm{L}}(v)$ can be fitted well with a ballistic form,

$$\lambda_{\mathrm{L}}(v) = \lambda_{\mathrm{L}} \left[ 1 - \left( \frac{v}{v_B} \right)^\nu \right], \tag{11}$$

as shown, for example, in Fig.14(a) for easy-plane anisotropy $\Delta = 0.3$ at $T = 0.96$. The deviation from the scaling for $v \gtrsim 1.5 v_B$ in the inset of Fig.14(a) and in Fig.5 is due to the numerical

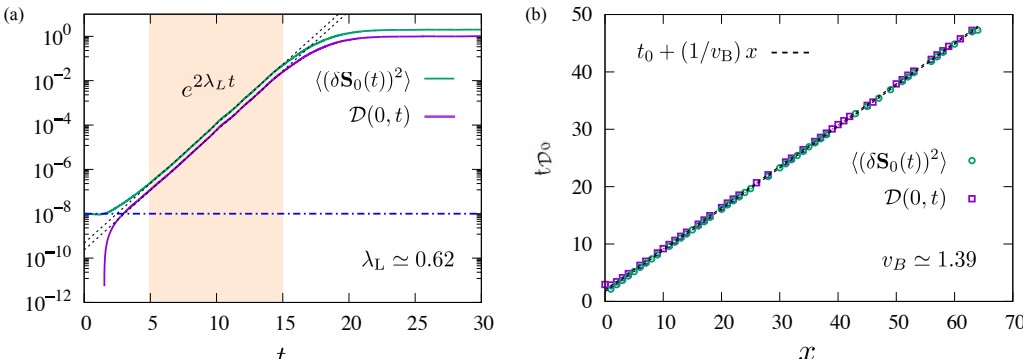

Figure 13: **Decorrelation and trajectory difference:** Comparison of decorrelation function $\mathcal{D}(x,t)$ and trajectory difference $(\langle\delta\mathbf{S}_x(t)\rangle^2)$ in calculation of (a) Lyapunov exponent and (b) butterfly velocity at temperature $T = 2.0$ for easy plane anistropy ($\Delta = 0.3$). From the exponential growth regime (shaded) in (a) we get same $\lambda_L \simeq 0.62$ and from inverse of the slope in (b) we found same $v_B$ from these two quantities.

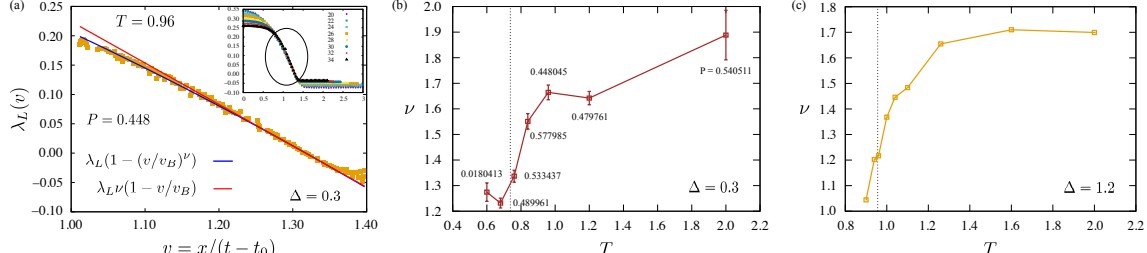

Figure 14: **Velocity-dependent Lyapunov exponent:** (a) Scaling collapsed of generalized Lyapunov exponent (zoomed in part inside the circle shown in the inset) with $v = x/(t-t_0)$ ($t > t_0$) at temperature $T = 0.96$ for $\Delta = 0.3$ (easy plane). Error bars are smaller than the data point symbols. We fit the region near zero crossing $v \sim 1.27 - 1.33$ with $\lambda_L(v) = \lambda_L \nu(1 - v/v_B)$ (red) for known $\lambda_L$ and $v_B$ and $\nu$ as fitting parameter. Non-linear function $\lambda_L(v) = \lambda_L(1 - (v/v_B)^\nu)$ for same values of the parameters is shown in blue. Goodness-of-fit indicator $P$-value is indicated. (b) Temperature dependence of $\nu$ extracted in the way mentioned in (a) for various temperature across the KT transition. $P$-values corresponding to each of the points are mentioned. (c) $\nu$ as a function of temperature for the easy-axis case across the Ising transition.

precision. The extracted values of the exponent $\nu$ are plotted as a function of $T$ for $\Delta = 0.3$ in Fig.14(b). Here to obtain $\nu$ we have used a linear approximation $\lambda_L(v) \simeq \lambda_L \nu(1 - v/v_B)$ to the non-linear fitting form in Eq.(11) for $v \sim v_B$ and fitted with $\nu$ as the fitting parameter for fixed values of $\lambda_L$ and $v_B$ obtained from Fig.3. The resulting linear fit and the non-linear function are compared with the data for $\Delta = 0.3$, $T = 0.96$ in Fig.14(a). We find $\nu \approx 1.9$ at high temperature, but $\nu$ decreases towards $\sim 1$ with decreasing temperature $T \gtrsim 0.6$. We could not get a reliable goodness of fit at lower temperature since the fitting range shrinks substantially for both $v > v_B$ and $v < v_B$ for $T \lesssim 0.6$.

To assess the goodness of the fits we obtain the errorbars [Fig.14(a)] in $\mathcal{D}(x,t)$ at each $(x,t)$ in terms of standard error in the mean (SEM), obtained by dividing $10^4$ trajectories generated with different initial conditions at temperature $T$ into multiple groups. Based on these errorbars on $\mathcal{D}(x,t)$ and $\chi^2$ fitting of the data for $\lambda_L(v)$ with the linear approximation to

Eq.(11) mentioned above, we obtain the error in $\nu$, and estimate the goodness of fit using [69]

$$P = 1 - \frac{1}{\Gamma(N_{\text{data}}/2)} \int_{\chi^2/2}^{\infty} y^{N_{\text{data}}/2-1} e^{-y} dy \,, \tag{12}$$

where $N_{\text{data}}$ is number of data points over the fitting range. A healthy fit is defined as $0.01 \lesssim P \lesssim P_{\text{max}}$, where $P_{\text{max}}$ is slightly less that 1 [69]. The errors in the estimate of $\nu$ and $P$ values are indicated in Fig.14(b) for the easy-plane case $\Delta = 0.3$. We have not carried out detail error analysis for easy-axis anisotropy $\Delta = 1.2$, but $\nu$ in this case is shown as a function $T$ in Fig.14(c).

To verify the possibility of the broadening of the chaos front around $\nu \approx \nu_B$, we have also tried to fit $\lambda_L(\nu)$ for $\nu \geq \nu_B$ with $\lambda_L(\nu) = \lambda_L(1 - (\nu/\nu_B))^{1+p}$ [34, 42] with $p$ as the fitting parameter. We find $p \simeq 0$, consistent with the absence of broadening [34] and the fact that the $\lambda_L(\nu)$ is more or less linear for $\nu > \nu_B$ [Figs.5,14(a)], over the range of $\nu$ where the scaling collapse works.

## D  Dynamical scaling law for OTOC

One can obtain dynamical scaling laws for OTOC and the butterfly speed $\nu_B$ across finite-temperature phase transitions with diverging length scale, as in the case of quantum critical point (QCP) [53]. To this end we consider

$$\mathcal{F}(\mathbf{r}, t) = 1 - \mathcal{D}(\mathbf{r}, t) = \langle \mathbf{S}_{a\mathbf{r}}(t) \cdot \mathbf{S}_{b\mathbf{r}}(t) \rangle \,. \tag{13}$$

We can write a scaling form for $\mathcal{F}(\mathbf{r}, t)$ by applying scaling transformation $\mathcal{F}(\mathbf{r}, t) = b^{-\Delta_{\mathcal{F}}} \Phi(L/b, \xi/b, r/b, b^{-z}t)$, where $z$ is the dynamical exponent, $\Phi$ a universal scaling function, and $\xi$ is the correlation length that diverges for $T \to T_c$ for Ising transition in the easy-axis case and for $T \leq T_{\text{KT}}$ at the KT transition and KT phase for easy-plane anisotropy. Since, $\mathcal{F}(\mathbf{r}, 0) = 1$ for any $L$, $\mathbf{r}$ and $\xi$, $\Delta_{\mathcal{F}} = 0$. Choosing $b = \xi$, we obtain

$$\mathcal{F}(\mathbf{r}, t) = \Phi(L/\xi, r/\xi, \xi^{-z}t) \,. \tag{14}$$

We consider two sets of parameters $(L_1, \xi_1, r_1)$ and $(L_2, \xi_2, r_2)$ such that $L_1/\xi_1 = L_2/\xi_2$, $r_1/\xi_1 = r_2/\xi_2$. At the scrambling time $t = t^*$, $\mathcal{F}(\mathbf{r}, t) = 1 - \varepsilon^2$, hence the scrambling times for the two sets of parameters are related by $\xi_1^{-z} t_1^* = \xi_2^{-z} t_2^*$. Thus the butterfly velocities $[\nu_B(L, \xi) = r/t^*]$ satisfy $\nu_B(L_1, \xi_1)\xi_1^{z-1} = \nu_B(L_2, \xi_2)\xi_2^{z-1}$. Based on this we can write down the scaling form for $\nu_B$

$$\nu_B(L, \xi) = \phi_B(\xi/L)\xi^{1-z} \,, \tag{15}$$

where $\phi_B$ is a scaling function. For $\xi/L \ll 1$, $\nu_B \sim \xi^{1-z}$. On the other hand, for finite $L$ and $\xi \to \infty$, $\phi_B(x) \sim x^{-(1-z)}$, implying $\nu_B \sim L^{1-z}$. These scaling forms are valid only for $z > 1$, since $\nu_B$ needs to be bounded due to causality.

## E  Spin auto-correlation functions

We calculate the spin auto-correlation functions $C_{ii}(t) = (1/N) \sum_{\mathbf{r}} \langle S_{\mathbf{r}}^i(t) S_{\mathbf{r}}^i(0) \rangle$ from the spin dynamics simulations starting with thermal initial conditions at different temperatures, as discussed in the main text. We mainly look into the planar correlation $C_{xy} \equiv C_{xx} + C_{yy}$ and out-of-plane correlation $C_{zz}$. More specifically, we compute

$$\tilde{C}_{ii}(t) = C_{ii}(t) - C_{ii}^{\infty} \,, \tag{16}$$

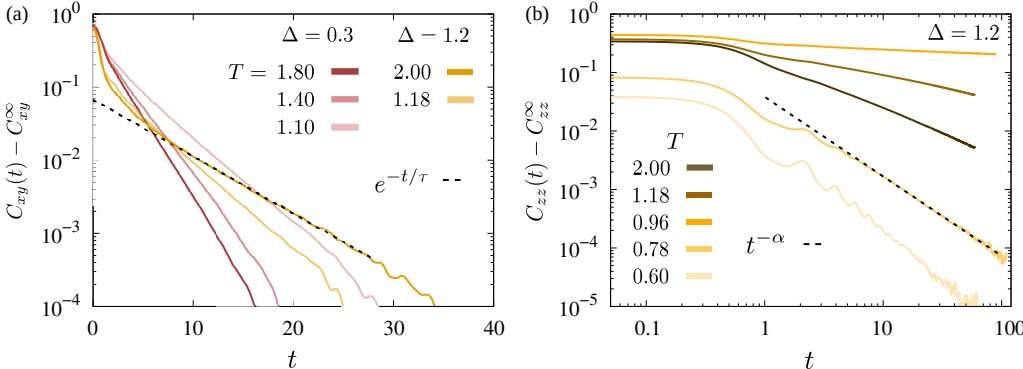

Figure 15: **Spin-spin auto-correlation function:** (a) Exponential decay of $\tilde{C}_{xy}(t)$ ($\sim e^{-t/\tau}$) for easy-plane and easy-axis anisotropies at temperature above the transitions. For $\Delta = 0.3$, the relaxation time ($\tau$) decreases with increasing temperature, whereas it increases for $\Delta = 1.2$ (see Figs.4(a),(b) in the main text). (b) Power law behaviour of $\tilde{C}_{zz}(t)$ ($\sim t^{-\alpha}$) at long times for $\Delta = 1.2$ across the Ising transition. $\alpha$ is found by calculating the slope of the linear regime ($t \gtrsim 5$) in the log-log plot, e.g. shown by the dashed line for $T = 0.78$. Similar plot for the easy-plane case is shown in Fig.6(a) of the main text.

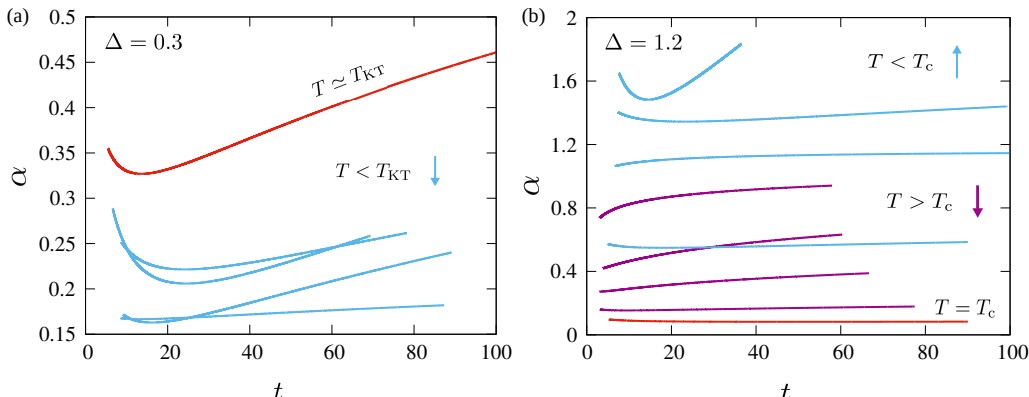

Figure 16: **Power-law exponent:** Time evolution of the local logarithmic slope $\alpha(t) = d\ln(C(t))/d\ln t$ for (a) $\Delta = 0.3$ and $T = 0.72, 0.68, 0.64, 0.60$ and $0.55$, where $C = \tilde{C}_{xy}$, and (b) $\Delta = 1.2$, across the Ising transition with $C = \tilde{C}_{zz}$ for $T = 2.00, 1.26, 1.10, 1.00, 0.96, 0.90, 0.82, 0.78$ and $0.74$. Different temperature regimes are marked with different colors. The direction of the arrows indicates the decrement of temperature.

where $C_{ii}^{\infty} = C_{ii}(t\to\infty) = (1/N)\sum_{\mathbf{r}}\langle S_{\mathbf{r}}^i(0)\rangle^2$, as we have $\langle S_{\mathbf{r}}^i(t\to\infty)S_{\mathbf{r}}^i(0)\rangle = \langle S_{\mathbf{r}}^i(t\to\infty)\rangle\langle S_{\mathbf{r}}^i(0)\rangle = \langle S_{\mathbf{r}}^i(0)\rangle^2$ for thermal initial conditions. As shown in Fig.9(a)(inset), in the easy-plane case, $C_{xy}^{\infty} \neq 0$ below $T_{\mathrm{KT}}$ since $\langle S_i^{x/y}(0)\rangle \neq 0$ for strong finite-size effect [64]. In the easy-axis case, $C_{zz}^{\infty} \neq 0$ below $T_c$ due to spontaneous symmetry breaking [Fig.9(b)(inset)].

In Fig.15(a), we show that $C_{xy}(t)$ decays exponentially with a relaxation time $\tau(T)$ for $T > T_{\mathrm{KT}}$ and $\Delta = 0.3$. Similar exponential decay is observed for the easy-axis case $\Delta = 1.2$. However, the relaxation time $\tau(T)$ increases approaching the transition for $\Delta = 0.3$, whereas it decreases for $\Delta = 1.2$, as shown in Figs.4(a),(b) in the main text. We show [Fig.15(b)] that $\tilde{C}_{zz}(t)$ exhibits a power-law decay for $t \gtrsim 5$ across the Ising transition. The exponent $\alpha$ changes from a diffusive value $\simeq 1$ at high temperature to sub-diffusive values ($< 1$) close to $T_c$, and finally to super-diffusive values ($> 1$) at low temperature [Fig.6(c)]. To verify whether

$\tilde{C}_{xy}(t)$ and $C_{zz}(t)$ have attained a steady power-law behaviour within our finite simulation time ($\lesssim 100$), we obtain a time-dependent exponent $\alpha(t) = d\ln(\tilde{C}(t))/d\ln t$ $(C = C_{xy}, C_{zz})$ for $T < T_{KT}$ in the easy-plane case [Fig.16(a)] and across $T_c$ for the easy-axis case [Fig.16(b)]. These suggest that a steady power-law exponent is achieve for $\Delta = 1.2$ except at the lowest temperature studied, whereas the exponent shows perceptible drift towards a larger value for $\Delta = 0.3$.

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
