# Peer review of "Many-body chaos and anomalous diffusion across thermal phase transitions in two dimensions"

_SciPost Physics, doi:SciPost Phys. 11, 087 (2021)_

## Round 2 · Referee Report · Anonymous (Referee 1) · 2021-2-13

Report

In this paper, Ruidas and Banerjee report spatio-temporal properties of dynamical correlation functions and chaos (suitably defined) in a classical spin model across thermal phase transitions and compare between the behaviours of chaos and long-time dynamics of collective modes. For the 2D classical XXZ model which harbours both Kosterlitz-Thouless and Ising transitions at different regimes of the anisotropy, they map out the dynamical properties in the different phases of the models using classical Monte-Carlo sampling and time evolution using the classical equations of motion. The underlying motivation is to explore connection between scrambling (or exponential decay of out-of-time-ordered correlation functions) as a diagnostic of chaos in quantum many body systems and the transport properties. However, in the absence of controlled numerical or analytical control over quantum systems of interest, they focus their study on a numerically tractable classical system, which could be viewed as the semi-classical limit of the quantum model. Their results suggest that, first, both chaos and transport show distinct features in the different thermodynamic phases. However, chaos follows a more universal ballistic behaviour across the phases, and the phase transition manifests only in the subtle form of crossover in the Lyapunov exponent and a minima in the butterfly velocity. Second, the dynamical auto-correlation function has sharper signatures of the phase transition, with diverging timescales and anomalous diffusion in the different phases, with distinct behaviour for KT and Ising transitions. Furthermore, their studies suggest that mechanisms of chaos and spin transport are different at low temperatures, and even at high temperature for the Ising case.

This study is a natural and necessary addition to the literature exploring the connection of many body chaos and transport, and more specifically to the more recent studies of butterfly effect in classical systems. The study also identifies several interesting features which would seed future studies in the field. Hence, I recommend this paper for publication, although I do have a few questions, concerns, and suggestions. I will list them out below. I hope that the authors address these issues before publication.

1. The authors consider two different measures of the butterfly effect in classical spin system, the decorrelator $D(x,t)$ and $\langle \delta S_{r}(t)^2\rangle$. All the reported results of the parameters such as Lyapunov exponent, $t_0$ and butterfly velocity are for the decorrelator, while it is claimed that the results are the same for the other quantity as well. I think this comparison warrants more evidence, if not in the main text, then as a section in the appendix. Also, why these quantities differ is not immediately clear from the text, and a little more exposition would be nice. Since the claim is that these quantities differ in $\mathcal{O}(\epsilon^2)$, and the data to be fitted are $\sim 10^{-8} \to 10^{-1}$ while $\epsilon \sim 10^{-4}$, there should be more clarity on how well the quantities derived from these two quantities match up.

2. The generalized Lyapunov exponent $D(x,t)$ is fitted to a ballistic form $\sim \exp\left[\lambda_L t(1-(x/v_{B}(t-t0))^\nu)\right] $, and it is claimed that $\nu \sim 2$ for all the cases, but there is no evidence shown for this, and it would be nice to see the goodness of the fit, especially as a function of Temperature. This is relevant, because in local quantum models, it has been shown that the generalized Lyapunov exponent is possibly not ballistic, but has a diffusively broadening wavefront, $\exp\left[\lambda_L t(1-(x/v_{B}(t-t0)))^{1+p}\right]$, with $p = 1$ definitively shown for random circuit models. From the locus of $\lambda_L(x,t) = 0$, there doesn’t seem to be broadening at these timescales in the model considered here, but it will be interesting the study this near the thermal phase transition, where, as the authors point out, chaotic behaviour could manifest similarly for classical and quantum case. I also think that the lack of wavefront broadening in their study is a significant finding and should be highlighted more.

3. For the easy plane case (KT transition), it is suggested that there is an anomalous subdiffusion for spin transport in the low temperature phase. However, as the authors also point out, the dynamical exponent $\alpha$ has an upward drift with time (Fig. 12 in appendix), so it is not clear whether this persists in the long-time limit. The authors claim that in these timescales, there is already evidence of ballistic chaotic behaviour, which would suggest that the mechanisms of dynamical spin transport and chaos are different. However, there is an onset time-scale for chaos, $t_0$ as denoted by the authors, which also seem to diverge at low temperatures. This would suggest a possible conflict, and the non-exponential behaviour of the decorrelator could coincide with subdiffusion at these timescales. Is this a possibility? If it is, then the authors should add a further qualification to the claims.

A few more minor points:

4. Fig 1 seems to have disconnected figures lumped together, which is disorienting in the first read. Fig 1a is perfect for the introduction, but it was not clear if the reader is supposed to pay attention to 1b, c, d, e while going through the introduction. I think it would improve the presentation of the paper if the 1 b-e is separated from the summary phase diagram and presented along with the technical results for chaos.

5. There are some issues with labelling in the figures. The authors consider two anisotropies - $\Delta = 0.3$ and $\Delta = 1.2$ - it will make the reader’s life much easier to label each relevant figure with this so that they are not disoriented. Furthermore, the legends in the plot should be labelled for clarity – for example, in 2c, the different plots are for temperatures, so that should appear in the label in the figures. Same for 1d, 2a, 2b, 3, 4a, 5a, 7, 9, 10 and 11.

---

## Round 2 · Referee Report · Anonymous (Referee 2) · 2021-3-10

Report

This manuscript analyzes various diagnostics of "chaos" for classical spin systems across two distinct thermal phase transitions. Overall, the manuscript provides a thorough and careful analysis of a set of questions that I am sure will be of interest to the wider community interested in many-body chaos in the quantum setting and its connection to the semiclassical limit. I recommend publication of this manuscript after the authors have addressed my comments and questions below. I also think that the authors should consider reorganizing their text, and especially their figures, such that it is easier for the readers to follow their discussion.

1) While the numerical analysis appears to be sound, I think the manuscript would benefit from a clearer discussion of how their results relate to the existing large-N computations discussed in Ref. [18] of the present manuscript (and to some extent, Ref. [41] of the quantum O(N) model studied within a different large-N setup). I realize that the authors have made some brief passing remarks, but I think the readers would benefit from a discussion of the key features that the earlier large-N calculations have clearly missed, which perhaps the present calculation captures better (For instance, I understand that the KT physics is much better captured here). The overall discussion is somewhat confusing and a bit haphazard, in my opinion.

2) Similarly, I am not sure to what extent the authors actually provide a rationalization of the temperature-dependence and the natural scale associated with the butterfly velocity (vB) away from the high-temperature limit (especially near the transition). Unlike in the quantum case, where theories with a well-defined "z" (dynamical exponent) show a temperature dependence that follows naturally, how does one understand the temperature dependence of vB in the present setting?

Minor comments:

1) On page 4, the authors write: "In contrast, short-range quantum models with finite local Hilbert space,....". I know that this has often been believed to be true,  but I know of at least one concrete "counterexample", i.e. a model with short-range interactions, a finite local Hilbert and without an obvious semiclassical limit involving random unitary circuits, which shows exponential growth of the OTOC over an extended period of time: arXiv:2009.10104. (The paper also explicitly presents the appropriate criterion that helps identify the regime where the exponential regime is well isolated.) For the sake of completeness, I think this should at least be cited.

2) On page 5, the authors while referring to the interplay of operator spreading in random unitary circuits and diffusion, make the remark that "these toy models are non chaotic". I am assuming that the authors have a very specific diagnostic in mind, related to the exponential growth of the OTOCs here? Otherwise, I think these models are still very much "chaotic" (though it might be hard to use many of the conventional metrics of chaos for these models). The authors should probably be more precise in their remark and also specify appropriate caveats.

---

## Round 3 · Author Response

We have revised our manuscript following your comments. We are appending our responses to Anonymous Report 1 and your comments below.

  1. Response to Anonymous Report 1

Comment 1: The authors have responded to all the questions raised in my report dili- gently and made the requisite changes. I recommend the article for publication. We thank the referee for useful comments and for going through our manuscript again.

Response: We thank the referee for recommending publication.

Comment 2: A minor point - some references are not up to date - it will be good to review the references and make sure that the latest references are used. One that I noticed was Ref 42 which is now published in Phys Rev B (https://journals.aps.org/prb/abstract/ 10.1103/Phys- RevB.102.184303).

Response: We have updated the above reference and other references with the citations to the latest published articles.

B. Response to Editor’s comment

Email Comment on 2021-7-28

We thank the editor for the important and useful comments. We have made modifications to the draft in response to the editor’s comments.

Comment 1: First, I think it would benefit from a pure English edit to break apart sentences with multiple subclauses, i.e. for examples. They make the introductory text very hard to parse.

Response: We have revised the introductory part following the above suggestion.

Comment 2: Second, I am somewhat confused about your use of the language of diffusion. The relevant conserved quantity is S z – yet in this manuscript you refer to C xy (t) ∼ 1/t a with small a in the KT phase as ‘sub-diffusion’. My understanding is that, if anything, the KT phase should exhibit superflow of S z , just like a U (1) breaking superfluid has j = ∇φ s super flow of n in parallel with a normal component of thermally activated phonons (the two fluid model). The latter would contribute a diffusive component to the transport of S z . There’s lots of work on spin superfluidity built on this picture. Am I missing some piece of physics and/or the way the language of diffusion is used in this context?

Response: We thank the editor for pointing out this important issue. We agree with the referee that it is not correct to attribute diffusion/subdiffusion to the power-law decay of C xy ∼ 1/t α , since the planar components are not conserved. Our intention was to contrast the power-law exponent α < 1 with ‘diffusive’ power-law. However, the use of subdiffu- sive/subdiffusion is indeed not legitimate. We have modified all the related texts. We also agree that there will be both superflow and diffusive parts to the transport of conserved component S z below T KT . Indeed, we find evidence of both spin-wave component, originating from ∇φ, and diffusive component in C zz (t), as expected from model E dynamics (Ref.60). As a result, C zz (t) exhibits an oscillatory behaviour with power-law decay having exponent consistent with α ≈ 1 . However, the power-law exponent cannot be detected accurately due to the oscillatory behaviour of C zz (t).

---

## Round 3 · List of Changes

We have made the following changes in response to the referee and editor’s comments.
1. We have modified the sentences with multiple sub-clauses to improve the readability of
the introduction part.
2. We have corrected all the sentences where the power-law decay of C xy (t) in the Kosterlitz-
Thouless regime for the easy-plane case was referred as ’subdiffusive’ and/or ‘subdiffu-
sion’.
3. We have updated the references with the citations to the latest published articles.

You are currently on this page

Resubmission 2007.12708v3 on 2 October 2021

---

## Editorial Decision

published